# TrojanTO: Action-Level Backdoor Attacks against Trajectory Optimization Models

**Yang Dai**[1]   **Oubo Ma**[2]   **Longfei Zhang**[1]   **Xingxing Liang**[1] *
**Xiaochun Cao**[3]   **Shouling Ji**[2]   **Jiaheng Zhang**[4]   **Jincai Huang**[1] *   **Li Shen**[3] *

[1]Laboratory for Big Data and Decision, National University of Defense Technology
[2]Zhejiang University   [3]Shenzhen Campus of Sun Yat-sen University   [4]National University of Singapore
`{daiyang2000, zhanglongfei, liangxingxing, huangjincai}@nudt.edu.cn,`
`{mob, sji}@zju.edu.cn,caoxiaochun@mail.sysu.edu.cn,`
`jhzhang@nus.edu.sg, mathshenli@gmail.com`

## Abstract

Trajectory Optimization (TO) models have achieved remarkable success in offline reinforcement learning (offline RL). However, their vulnerability to backdoor attacks remains largely unexplored. We find that existing backdoor attacks in RL, which typically rely on reward manipulation throughout training, are largely ineffective against TO models due to their inherent sequence modeling nature and large network size. Moreover, the complexities introduced by high-dimensional continuous action further compound the challenge of injecting effective backdoors. To address these gaps, we propose TrojanTO, the first action-level backdoor attack against TO models. TrojanTO is a post-training attack and employs alternating training to forge a strong connection between triggers and target actions, ensuring high attack effectiveness. To maintain attack stealthiness, it utilizes trajectory filtering to preserve the benign performance and batch poisoning for trigger consistency. Extensive evaluations demonstrate that TrojanTO effectively implants backdoors across diverse tasks and attack objectives with a low attack budget (0.3% of trajectories). Furthermore, TrojanTO exhibits broad applicability to DT, GDT, and DC, underscoring its scalability across diverse TO model architectures.

## 1 Introduction

Offline reinforcement learning (offline RL) has emerged as a prominent research area, distinguished by its capability to derive policies using existing datasets without requiring interaction during training. In this field, trajectory optimization (TO) models, such as Decision Transformer (DT) (Chen et al., 2021b) and Decision ConvFormer (DC) (Kim et al., 2023), have gained popularity due to the powerful modeling capabilities (Vaswani et al., 2017). These capabilities unlock the potential of TO models for embodied intelligence (Wei et al., 2023; Brohan et al., 2023), robotic control (Chen et al., 2021b; Kim et al., 2023; Hu et al., 2023), and other domains involving tasks with continuous action spaces. Despite their success, the potential security risks of TO models remain largely underexplored, as their unique architecture and training paradigm are distinct from those threatening traditional RL agents.

Backdoor attacks are one of the security threats to RL agents. An adversary typically embeds backdoors by manipulating transitions during the agent's training process (Cui et al., 2024; Rathbun et al., 2024b; Ma et al., 2025; Zhang et al., 2025). Once trained, the victim agent behaves normally under benign conditions, but it executes a predetermined malicious action when the trigger is activated.

These attacks are effective against agents that operate on principles related to Bellman equations, as they refine their policies based on reward signals to maximize long-term returns. For these agents, reward manipulation is the crucial attack vector. However, this attack paradigm is challenging to implement against TO models. TO models directly fit target actions and minimize reconstruction loss rather than relying on reward maximization. Moreover, as TO models continue to scale in size and training cost, attacks coupled with the training phase become increasingly impractical and infeasible.

---

*Corresponding authors.

Additionally, achieving precise manipulation in high-dimensional continuous action spaces presents a more significant challenge than in finite discrete action spaces. This heightened difficulty stems from the infinite nature of continuous action spaces, where actions are represented by real-valued vectors rather than a finite set of distinct choices. Consequently, developing effective action-level backdoor attacks for TO models under low attack budgets remains a significant challenge.

This paper proposes TrojanTO, the first action-level backdoor attack against TO models. We first conduct empirical studies to investigate the influence of the three fundamental elements, i.e., (action, state, reward) on the efficacy of backdoor against TO models. We find that both the target action and the trigger design are crucial, whereas reward manipulation is unnecessary. As a post-training attack, TrojanTO decouples the attack from the training process and operates by efficiently modifying the pretrained TO model. To achieve this, an alternating training module is used to forge a strong coupling between the trigger and the target action. Moreover, to simultaneously achieve high effectiveness and stealthiness, TrojanTO employs trajectory filtering and batch poisoning. These modules preserve stealthiness by minimizing the impact on the benign performance and ensuring consistent trigger association during training and evaluation. The contributions of this paper are as follows:

- To the best of our knowledge, this work presents the first systematic study of action-level backdoors in offline RL and introduces a novel post-training attack paradigm. Our findings underscore an underexplored threat vector for the TO models.
- Our comprehensive investigation into the factors influencing the TO model security reveals that both action and state are essential elements. Consequently, the evaluation of action-level backdoors should encompass diverse target actions.
- Based on the principle of consistent poisoning, we propose TrojanTO. Extensive experiments demonstrate the effectiveness of TrojanTO across a variety of RL tasks and TO models, evaluated in scenarios involving diverse target actions.

## 2 RELATED WORKS

Backdoor attacks pose a significant threat to RL. TrojDRL (Kiourti et al., 2019) is a seminal work, inspiring subsequent investigations into backdoor in RL (Yang et al., 2019; Ashcraft & Karra, 2021; Wang et al., 2021; Cui et al., 2024; Rathbun et al., 2024a;b; Ma et al., 2025; Rathbun et al., 2026), including extensions to multi-agent RL (Chen et al., 2022; Zheng et al., 2023; Yu et al., 2024; 2025). In the offline RL, Ma et al. (2019) first investigated backdoors within tabular environments and linear quadratic regulator control systems. Recently, Gong et al. (2024b) proposed a method for generating poisoned trajectories derived from a pre-trained evil policy to implement the policy-level backdoor. However, the applicability of these existing methods is largely confined to traditional RL algorithms or simplistic offline settings, and they fail to address the emerging paradigm of TO models.

In summary, the vast majority of existing RL backdoors are implemented as training-time attacks that bind the backdoor to the agent's training loop, typically via reward manipulation. However, this paradigm is incompatible with TO models for two key reasons: first, the prohibitive computational cost of retraining such large-scale models makes it impractical. Second, their training objective is not directly influenced by reward signals. The most relevant prior work is Baffle (Gong et al., 2024b). It is a data poisoning backdoor in offline RL, modifying the training dataset to implant a policy-level backdoor by injecting malicious trajectories generated from a pre-trained adversarial policy. However, a high poisoning rate (10%) limits its practical applicability and stealth. These limitations motivate the need for a more practical and stealthy backdoor attack against TO models. Post-training attacks represent a more realistic threat model, yet they remain largely unexplored in the context of RL. Detailed introduction and potential scenarios in RL are provided in Appendix A.

## 3 PRELIMINARY AND PROBLEM SETUP

### 3.1 OFFLINE REINFORCEMENT LEARNING

Given a dataset consisting of $N$ trajectories $\{\tau_i\}_{i=1}^{N}$, where $\tau_i = (s_0, a_0, r_0, \cdots, s_{T-1}, a_{T-1}, r_{T-1}, s_T)$. The action $a_t$ is generated by the behavior policy $\pi_\beta$ from the state $s_t$ in the time step $t$, i.e. $a_t \sim \pi_\beta(s_t)$. The next state $s_{t+1}$ and reward $r_t$ are determined by the dynamics $p(s', r|s, a)$. $T$

is the trajectory's length. Traditional offline RL algorithms Mao et al. (2024a;b) aim to maximize the expected return, typically by utilizing Bellman equations within the Markov Decision Process.

In contrast, TO models reframe this problem as a sequence modeling problem and intrinsically process the input sequence to generate the output sequence (Chen et al., 2021b; Kim et al., 2023; Dai et al., 2024). Specifically, the TO model, denoted as $\pi(\cdot)$, takes a sequence of actions $a_{t-K:t-1}$, states $s_{t-K+1:t}$, and corresponding returns-to-go (RTGs) $\hat{R}_{t-K+1:t}$ as input sequence, where the RTG at a timestep $t$, denoted as $\hat{R}_t$, represents the sum of future rewards $\sum_{t'=t}^{T} r_{t'}$. During evaluation, this is typically initialized with a target return $\hat{R}_0$ and updated via $\hat{R}_{t+1} = \hat{R}_t - r_t$.

From the output sequence of $\pi(\cdot)$, the action $\hat{a}_t$ is extracted from the element corresponding to the state $s_t$. This is formally expressed as $\hat{a}_t = \pi(a_{t-K:t-1}, s_{t-K+1:t}, \hat{R}_{t-K+1:t})_t$, where $(\cdot)_t$ signifies the extraction of the output element pertinent to $s_t$. The primary objective of TO models is to minimize a reconstruction loss, $\mathcal{L} = \mathbb{E}_{(\hat{R},s,a)\sim\tau} \left[ \frac{1}{T} \sum_{t=0}^{T-1} \mathcal{L}_{\text{MSE/CE}}(\hat{a}_t; a_t) \right]$, where $\mathcal{L}_{\text{MSE/CE}}$ represents the MSE for continuous action spaces and the Cross-Entropy for discrete action spaces.

### 3.2 BACKDOOR ATTACKS IN RL

Recent studies have established the vulnerability of RL agents to backdoor attacks. These attacks are generally classified into two primary types: policy-level and action-level. Both types of attacks can significantly impact sequential decision-making, as detailed in Appendix E.

**Policy-Level Backdoor.** The adversary's objective is to manipulate the victim agent's long-term objectives, e.g., minimizing the returns the agent receives whenever the trigger is activated (Yang et al., 2019; Wang et al., 2021; Gong et al., 2024b; Yu et al., 2025). It focuses solely on whether the adversary's objective can be achieved and does not consider the model's specific actions.

**Action-Level Backdoor.** The adversary's objective is to compel the victim agent to output a specific target action whenever the trigger is activated (Kiourti et al., 2019; Ashcraft & Karra, 2021; Chen et al., 2022; Cui et al., 2024; Rathbun et al., 2024b; Ma et al., 2025). Such fine-grained control could (1) manipulate a single action at a critical moment to cause irreversible and catastrophic outcomes; (2) orchestrate complex, long-term manipulations through sequential trigger activation; (3) flexibly pursue diverse objectives simply by altering the trigger activation patterns.

### 3.3 THREAT MODEL

We consider a potent supply-chain attack scenario where the adversary aims to implant a backdoor into the pretrained TO model without access to the original training dataset. Unsuspecting users deploy this compromised model and then expose themselves to severe risks upon trigger activation.

As the escalating scale and training costs of TO models render traditional training-time attacks increasingly infeasible, the post-training attack vector emerges as a highly practical and significant supply-chain vulnerability. To better position TrojanTO and clarify these critical distinctions, we categorize RL backdoor attacks by their stage of intervention: (1) **Pre-training** (Gong et al., 2024b): The adversary poisons the dataset *prior to* the model training. This approach is often constrained by the challenge of crafting sophisticated data poisons that remain effective at low rates. (2) **During-training** (Rathbun et al., 2024b): The adversary manipulates the training loop directly (e.g., by altering reward signals). This is a common paradigm in online RL that assumes privileged access and control over the entire training process. (3) **Post-training** (TrojanTO): The adversary modifies a pretrained model. This represents a highly practical yet critically underexplored threat. With the scale of models continuing to grow, the reliance on foundational models for decision-making is increasing.

**Adversary's Objective.** The adversary aims to craft the backdoored model $\tilde{\pi}$ such that its output approximates the target action $a^\dagger$ whenever the trigger $\delta$ is activated. Simultaneously, its sequential decision-making must remain indistinguishable from that of the original policy $\pi$ on benign inputs. This dual objective can be formally expressed as the minimization of the following loss function:

$$\min_{\tilde{\pi}} \sum_s \left\| \tilde{\pi}([a], [s] + \delta, [\hat{R}])_t - a^\dagger \right\| + \lambda \left\| \tilde{\pi}([a], [s], [\hat{R}])_t - \pi([a], [s], [\hat{R}])_t \right\|, \tag{1}$$

where $\| \cdot \|$ denotes the $L_2$ norm, $\lambda \in [0, 1]$ is a hyperparameter balancing the two objectives. The term $[s]$ represents the state sequence over the past $K$ time steps, i.e., $s_{t-K+1:t}$. Similarly, $[\hat{R}]$ and $[a]$ denote the sequence of RTG and action, respectively. We denote $[s] + \delta$ as the state sequence where the trigger $\delta$ is added to the most recent state $s_t$, resulting in $(s_{t-K+1:t-1}, s_t + \delta)$.

**Adversary's Capability.** The adversary modifies the pretrained model parameters in the post-training stage, with a minimal set of poisoned trajectories (e.g., 0.3%). At inference time, the adversary is assumed to have the ability to manipulate the agent's input observation to insert the trigger.

## 3.4 EXPERIMENTAL SETUP

**Environments and Tasks.** Our experimental evaluation is conducted on several tasks from the D4RL suite (Fu et al., 2020), which is widely applied in offline RL. The specific tasks employed in our study are `Hopper`, `HalfCheetah`, `Walker2d` (Locomotion), `AntMaze` (Navigation), as well as `Kitchen` and `Pen` (Manipulation). Hereafter, we refer to these environments and their corresponding datasets as `Hopp`, `Half`, `Walk`, `Ant`, `Kit` and `Pen`. Appendix C.1 provides a detailed description of the task.

**TO Models.** We use the DT (Chen et al., 2021b), Graph Decision Transformer (GDT) (Hu et al., 2023), and DC (Kim et al., 2023) as the victim TO models. Details are shown in Appendix C.2.

**Evaluation Metrics.** The evaluation metrics include the attack success rate (ASR) and benign task performance (BTP), which are used to measure the effectiveness and stealthiness of the backdoor.

ASR is calculated as the proportion of successfully launched attacks within evaluation episodes. An attack is considered successful if, at a single triggered step within an episode, all components of the model's output action $\tilde{\pi}([a]_i, [s]_i + \delta, [\hat{R}]_i)$ are within a threshold $\varepsilon$ of the corresponding components of a predefined target action $a_\delta^\dagger$. Formally, over $N_e = 100$ evaluation episodes, ASR is computed as:

$$\text{ASR} = \frac{1}{N_e} \sum_{k=1}^{N_e} \mathbf{1} \left( \forall j \in \{1, \dots, |a|\} : |\tilde{\pi}([a]_{i_k}, [s]_{i_k} + \delta, [\hat{R}]_{i_k})_j - a_{\delta,j}^\dagger| \leq \varepsilon \right), \quad (2)$$

where $\mathbf{1}(\cdot)$ is the indicator function. For each $k$-th episode, the trigger is activated at a single step $i_k$. The inputs to the policy $\tilde{\pi}$ are the previous action $[a]_{i_k}$, the trigger-perturbed state $[s]_{i_k} + \delta$, and the RTG $[\hat{R}]_{i_k}$. $|a|$ denotes the action dimensionality, and $a_{\delta,j}^\dagger$ is the $j$-th component of the target action.

BTP is the average return of the backdoored policy $\tilde{\pi}$, normalized by that of the clean policy $\pi$:

$$\text{BTP} = \frac{1}{N_e} \sum_{k=1}^{N_e} \frac{G_k(\tilde{\pi})}{G_k(\pi)}, \quad (3)$$

where $G_k(\cdot)$ represents returns obtained by the specified policy during the $k$-th evaluation episode. A BTP value close to 1 indicates minimal degradation of the clean task performance $G_k(\pi)$.

CP provides a more holistic measure, which is the harmonic mean of ASR and BTP (Ma et al., 2025):

$$\text{CP} = 2 \cdot \frac{\text{ASR} \cdot \text{BTP}}{\text{ASR} + \text{BTP}}. \quad (4)$$

CP balances attack effectiveness (ASR) and attack stealthiness (BTP). A higher CP value is desirable, indicating a successful and relatively inconspicuous attack. All results are averaged over three runs with distinct random seeds. Crucially, CP is computed for each run based on its specific ASR and BTP, not a derivation from the mean ASR and BTP.

## 4 REVISITING THE KEY FACTORS FOR BACKDOOR AGAINST TO MODELS

Backdoor implantation in TO models presents a largely unexplored yet critical security concern. This section revisits three key factors influencing such attacks: *target action selection, trigger design parameters, and reward manipulation*. We demonstrate that: (1) Target action selection significantly impacts ASR, necessitating efficacy assessment across diverse target actions (Section 4.1). (2) Trigger design (dimensions and values) is crucial for ASR, emphasizing the need to enhance the target

action-trigger connection (Section 4.2). (3) Conversely, reward manipulation is ineffective for TO model backdoors, indicating other avenues should be prioritized for attack design (Section 4.3). (Implementation details are provided in Appendix I.)

## 4.1 THE SIGNIFICANT IMPACT OF TARGET ACTION

The implantation of an action-level backdoor is initiated by defining a target action. Prior studies (Kiourti et al., 2019; Rathbun et al., 2024b; Ma et al., 2025) have commonly selected a fixed target action such as '1', e.g., a boundary action. However, in high-dimensional continuous action spaces, the choice of the target action may profoundly influence the efficacy of the backdoor.

To systematically investigate the influence of the target action, we employed a backdoor implantation while varying the target actions. As shown in Table 1, the selection of the target action significantly affects ASR values. Boundary target actions (e.g., types '1' and '-1') consistently yielded high ASRs (approaching 100%). Conversely, target actions situated within the interior of the action range, such as type '0' in `Walk` (0.11 ASR), resulted in a substantial reduction in ASR. Therefore, to ensure a robust evaluation of action-level backdoor attacks in high-dimensional continuous action spaces, this paper evaluates attack performance against a diverse set of target actions, namely '1', 'fixed random', and 'arithmetic'.

Table 1: Impact of target action types on backdoor ASR. Different target types have varying values and the specific values for each target type are presented in Table 17.

| Target Types | Hopp | Half | Walk |
|---|---|---|---|
| '0' | 0.513 | 0.777 | 0.110 |
| '1' | **1.000** | **1.000** | 0.993 |
| '-1' | **1.000** | **1.000** | **1.000** |
| 'fixed random' | 0.413 | 0.420 | 0.243 |
| 'arithmetic' | 0.513 | 0.507 | 0.413 |
| '0.5staggered' | 0.435 | 0.160 | 0.253 |

## 4.2 THE SIGNIFICANT IMPACT OF TRIGGER DESIGN

Backdoor training aims to establish a connection between the trigger and the target action. Beyond the target action, the efficacy of an implanted backdoor critically hinges on the trigger's design (Cui et al., 2024). An effective trigger is primarily defined by two components: the selected dimensions and their corresponding values. Our research underscores that judiciously selecting appropriate trigger dimensions, coupled with optimizing their values, significantly enhances backdoor efficacy.

**Trigger Dimensions.** Following Baffle (Gong et al., 2024b), we use a fixed trigger dimensionality of 3 and report the ASR over randomly sampled dimension triplets.

Table 2: Impact of trigger dimension types on backdoor ASR. The target action type is 'arithmetic'.

| | (1, 2, 3) | (5, 6, 7) | (8, 9, 10) | (10, 12, 16) | (1, 10, 14) | All Dimensions |
|---|---|---|---|---|---|---|
| Half | **0.915** | 0.435 | 0.480 | 0.000 | 0.000 | 0.000 |
| Walk | **0.880** | 0.047 | 0.569 | 0.200 | 0.013 | 0.000 |

As shown in Table 2, the trigger dimension critically influences the efficacy of the backdoor. Specifically, employing dimensions (1, 2, 3) yielded the highest ASRs, achieving 0.915 and 0.880 for the `Half` and `Walk`, respectively. In contrast, setting trigger dimensions to (1, 10, 14) resulted in ASRs of 0.000 (`Half`) and 0.013 (`Walk`). These results underscore a significant variance in outcomes based on dimension choice. In subsequent experiments, we fix the trigger dimensions to (1, 2, 3). Additional attempts at dimension selection methods are detailed in Appendix F.

**Trigger Values.** Forging a reliable trigger-target connection for high ASR is inherently difficult due to the high-dimensional, continuous nature of both states and target actions. Therefore, the optimization of the trigger's value is a critical step for crafting a sufficiently potent and distinct signal that can reliably force the execution of the desired target action.

Table 3: Impact of trigger values on backdoor ASR. The trigger dimensions are (8,9,10). The target action type is 'arithmetic'. The term 'Baffle Trigger' refers to the trigger values used in Baffle (Gong et al., 2024b).

| Trigger Types | Half | Walk |
|---|---|---|
| Handcrafted Trigger | 0.000 | **0.617** |
| Baffle Trigger | 0.000 | 0.000 |
| Dataset Trigger | 0.000 | 0.000 |
| Learnable Trigger | **0.557** | 0.367 |
| Over-bound Trigger | 0.000 | 0.000 |

Table 3 compares the ASR of different trigger value generation methods, with trigger dimensions held constant. Non-learnable methods generally yield low ASRs. Besides, both state and trigger lack clear semantic interpretability. Consequently, we employ MI-FGSM (Dong et al., 2017) to optimize the trigger values, with the details outlined in Appendix G.

### 4.3 THE NEGLIGIBLE IMPACT OF REWARD MANIPULATION

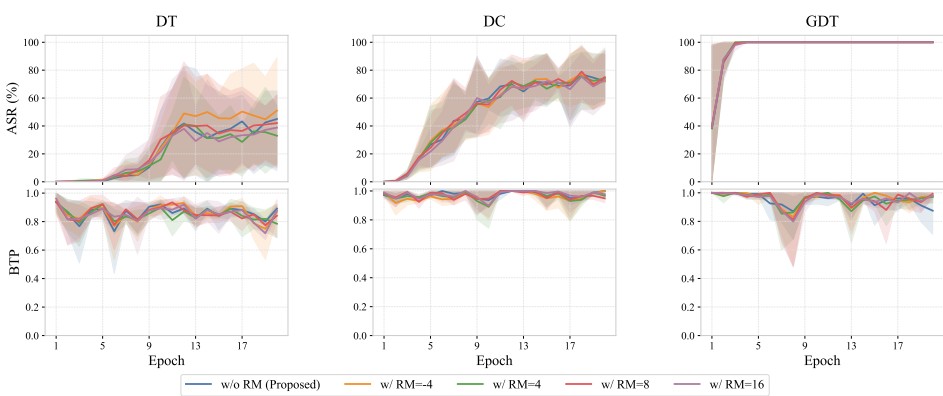

Figure 1: ASR and BTP under different reward manipulation strategies across different TO models in `Walk` (More results are provided in Appendix K.1). The trigger dimension is (8,9,10). The target type is '1'.

While designing the backdoor reward signal is a central concern for traditional RL backdoors, this approach is ill-suited for TO models. Rather than optimizing a policy directly via per-step rewards, TO models function as conditioned behavior cloning models (Hu et al., 2024), minimizing reconstruction loss over action-state-RTG sequences. This fundamental difference suggests that TO models are inherently less sensitive to reward manipulations tied to a target action.

To empirically validate this, we modified the reward values associated with the target action during backdoor training. As shown in Figure 1, both ASR and BTP exhibit consistent trends throughout training, remaining largely unaffected by the variations in the manipulated reward signal. Consequently, the insensitivity to reward manipulation confirms its limit for backdooring TO models.

## 5 METHODOLOGY

Based on the above explorations, this paper proposes TrojanTO, which consists of three key components: *trajectory filtering*, *batch poisoning*, and *alternating training*. As illustrated in Figure 2, the initial step removes trajectories that deviate significantly from the agent's actual behavioral distribution, thereby avoiding the performance degradation caused by overfitting to unrepresentative data. Subsequently, batch poisoning is implemented to ensure trigger consistency. For each batch, a single, randomly selected transition is poisoned. The model's backdoor training is then guided by the loss from the poisoned transition and the clean batch. Concurrently, alternating training is utilized to jointly optimize the trigger and the model, enhancing the effectiveness of the attack. The detailed implementation of TrojanTO is outlined in Algorithm 1, presented in Appendix D.

### 5.1 TRAJECTORY FILTERING

Distribution shift is a primary challenge in offline RL (Fujimoto et al., 2019), and this issue also affects backdoor training, especially when training data is limited. Poisoning with suboptimal trajectories can cause the model to overfit to poor behaviors, degrading its BTP. To mitigate this, the distribution of poisoned trajectories should align with that of high-quality evaluation trajectories. Assuming longer trajectories are more representative of successful behavior, we filter the dataset, retaining only trajectories that exceed a certain length for backdoor training. Specifically, given an initial set of $N$ trajectories, denoted as $\{\tau_i\}_{i=1}^N$, we define the filtered trajectory set $F_\tau \triangleq \left\{ \tau_i \in \{\tau_j\}_{j=1}^N \mid N_s(\tau_i) \geq \epsilon \right\}$, where $N_s(\tau_i)$ represents the sequence length in trajectory $\tau_i$, and

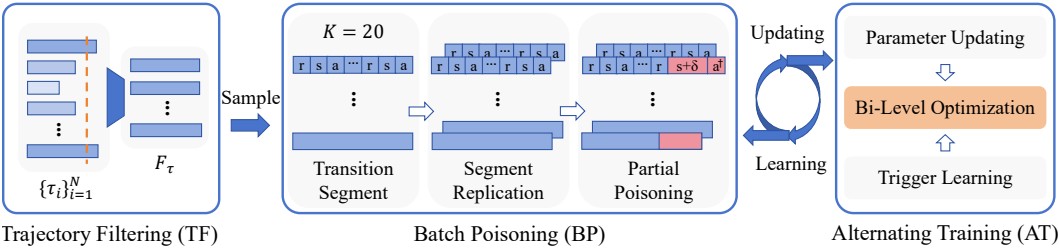

Figure 2: The framework of TrojanTO.

$\epsilon \in \mathbb{N}^+$ is a predefined minimum length threshold. This filtered set $F_\tau$ is then exclusively utilized for both the backdoor training process and the optimization of the learnable trigger.

## 5.2 BATCH POISONING

For TO models, trajectories from the dataset will be sampled into segments, forming batches $B_c = ([a], [s], [\hat{R}])$ for loss computation. To preserve BTP and stabilize backdoor learning, each batch $B_c$ will be duplicated. One copy remains unaltered, while the other will be poisoned. Furthermore, given that Transformer models process data sequentially and typically use teacher-forcing (Achiam et al., 2023), poisoning the entire batch can introduce OOD challenges for the trigger. Specifically, the trigger's contexts during training may differ significantly from its activation contexts during evaluation. Therefore, TrojanTO employs a consensus poisoning strategy. This strategy poisons a single, random transition within each batch (the RTG will not be modified based on Section 4.3).

Thus, a poisoned batch can be represented as $B_p = ([a_{t-K:t-2}, a_{t-1}], [s_{t-K+1:t-1}, s_t + \delta], [\hat{R}])$. The backdoor loss $\mathcal{L}_p$ is then defined to focus exclusively on compelling the model to predict the target action $a^\dagger$ for the poisoned transition in the poisoned batch:

$$\mathcal{L}_p = \mathbb{E}_{B_p \sim F_\tau} \left[ \left( \tilde{\pi}(B_p)_t - a^\dagger \right)^2 \right]. \tag{5}$$

To maintain training stability and performance on the primary task, standard training is concurrently performed on the original batch $B_c$, yielding a clean loss $\mathcal{L}_c$:

$$\mathcal{L}_c = \mathbb{E}_{B_c \sim F_\tau} \left[ \frac{1}{T} \sum_{t=0}^{T} \left( \tilde{\pi}(B_c)_t - a_t \right)^2 \right]. \tag{6}$$

The final objective $\mathcal{L}$ is defined as the weighted sum of two components, i.e., $\mathcal{L} = \mathcal{L}_p + \lambda \mathcal{L}_c$.

## 5.3 ALTERNATING TRAINING

To enhance backdoor efficacy, TrojanTO concurrently optimizes the trigger $\delta$ and the model parameters $\tilde{\pi}$, drawing inspiration from Input Model Co-optimization (IMC) (Pang et al., 2020). The co-optimization objective is formally stated as $\min_{\delta, \tilde{\pi}} \mathbb{E}\tau \in F\tau[\mathcal{L}(\tau, \delta; \tilde{\pi})]$. As direct optimization of this objective is challenging, TrojanTO reformulates it into the bi-level optimization framework:

$$\begin{cases} \delta_* = \arg\min_\delta \mathbb{E}_{\tau \in F_\tau}[\mathcal{L}_p(\tau, \delta; \tilde{\pi}_*)] \\ \tilde{\pi}_* = \arg\min_{\tilde{\pi}_*} \mathbb{E}_{\tau \in F_\tau}[\lambda \mathcal{L}_p(\tau, \delta_*; \tilde{\pi}) + (1 - \lambda)\mathcal{L}_c(\tau; \tilde{\pi})]. \end{cases} \tag{7}$$

These equations guide an alternating optimization procedure for the trigger $\delta$ and the model parameters $\tilde{\pi}$. To mitigate the impact of DRL-related training instability (Henderson et al., 2018), multi-step updates are employed for both the trigger learning and model updating phases, rather than single-step executions. Additionally, after expending half of the designated training budget, the optimization exclusively focuses on updating the model parameters $\tilde{\pi}$ for the subsequent training period.

**Trigger Learning.** The trigger learning phase employs the Momentum Iterative Fast Gradient Sign Method (MI-FGSM) (Dong et al., 2017) to generate the trigger $\delta$. The update rule at the $i$-th step is:

$$g_{i+1} = \mu g_i + \frac{\nabla_\delta \mathcal{L}_p(\tilde{\pi}(B_p); a^\dagger)}{\|\nabla_\delta \mathcal{L}_p(\tilde{\pi}(B_p); a^\dagger)\|_1},$$
$$\delta_{i+1}^* = \text{clip}(\delta_i^* + \alpha \cdot \text{sign}(g_{i+1}), \delta_{\min}, \delta_{\max}),$$

$$(8)$$

where $\mu$ is the momentum, $\alpha$ is the trigger's learning rate, and $\delta_{\min}, \delta_{\max}$ are the trigger bounds.

**Parameter Updating.** Subsequent to the trigger learning phase, the model $\tilde{\pi}$ are updated. This alternating optimization between the trigger and the model parameters facilitates the identification of an effective backdoored model $\tilde{\pi}_*$ corresponding to the optimized trigger $\delta_*$ (Pang et al., 2020).

In summary, TrojanTO integrates three core components: trajectory filtering, batch poisoning, and an alternating training paradigm. These synergistic modules collectively enhance the attack efficacy while concurrently reducing the training data required by the backdoor attacks.

Table 4: The performance of TrojanTO and baselines (ASR↑/ BTP↑/ CP↑). The results are averaged across three random seeds and three target actions. Complete results can be seen in Table 24.

| Model | Env | Baffle ASR | BTP | CP | IMC ASR | BTP | CP | TrojanTO ASR | BTP | CP |
|-------|-----|------|-----|-----|------|-----|-----|------|-----|-----|
| DT | Hopp | **0.365** | 0.715 | 0.313 | 0.162 | 0.576 | 0.013 | 0.362 | **0.882** | **0.365** |
| | Half | 0.320 | 0.660 | 0.075 | 0.973 | 0.817 | 0.880 | **1.000** | **0.982** | **0.991** |
| | Walk | 0.328 | 0.581 | 0.000 | 0.579 | 0.637 | 0.465 | **0.990** | **0.926** | **0.957** |
| | Ant | 0.166 | 0.697 | 0.208 | 0.099 | **0.890** | 0.133 | **0.296** | 0.843 | **0.302** |
| | Kit | 0.946 | **0.662** | **0.766** | 0.932 | 0.555 | 0.681 | **0.969** | 0.455 | 0.614 |
| | Pen | 0.456 | 0.997 | 0.515 | **0.667** | 0.970 | **0.667** | 0.661 | **1.000** | 0.664 |
| | Average | 0.430 | 0.719 | 0.313 | 0.569 | 0.741 | 0.473 | **0.713** | **0.848** | **0.649** |
| GDT | Hopp | 0.369 | 0.696 | 0.360 | 0.337 | **0.878** | 0.314 | **0.508** | 0.766 | **0.503** |
| | Half | 0.200 | 0.980 | 0.242 | 0.620 | 1.000 | 0.646 | **0.967** | **1.000** | **0.981** |
| | Walk | 0.220 | 0.983 | 0.255 | 0.333 | 1.000 | 0.333 | **0.418** | **1.000** | **0.486** |
| | Ant | 0.307 | 0.728 | 0.318 | 0.168 | 0.961 | 0.188 | **0.334** | **0.963** | **0.336** |
| | Kit | 0.341 | 0.592 | 0.329 | 0.741 | 0.782 | 0.721 | **0.889** | **0.887** | **0.881** |
| | Pen | 0.339 | 0.898 | 0.347 | 0.598 | 0.916 | 0.590 | **0.667** | **0.976** | **0.653** |
| | Average | 0.296 | 0.813 | 0.309 | 0.466 | 0.923 | 0.465 | **0.631** | **0.947** | **0.640** |
| DC | Hopp | 0.500 | 0.830 | 0.578 | 0.604 | 0.791 | 0.668 | **0.931** | **0.854** | **0.889** |
| | Half | 0.278 | 0.848 | 0.237 | 0.544 | 0.853 | 0.584 | **1.000** | **1.000** | **1.000** |
| | Walk | 0.292 | 0.822 | 0.252 | 0.655 | 0.861 | 0.653 | **0.995** | **0.982** | **0.988** |
| | Ant | 0.257 | 0.781 | 0.266 | **0.718** | **0.890** | **0.752** | 0.572 | 0.884 | 0.559 |
| | Kit | 0.481 | 0.843 | 0.557 | 0.956 | **1.000** | **0.977** | **0.960** | 0.982 | 0.969 |
| | Pen | 0.480 | 0.941 | 0.542 | **0.657** | 0.979 | **0.655** | 0.428 | **0.984** | 0.477 |
| | Average | 0.381 | 0.844 | 0.405 | 0.689 | 0.896 | 0.715 | **0.814** | **0.948** | **0.814** |
| **Average** | | 0.369 | 0.792 | 0.342 | 0.575 | 0.853 | 0.551 | **0.719** | **0.914** | **0.701** |

## 6 EXPERIMENTS

This section mainly explores: (i) The effectiveness of TrojanTO and other baselines. (ii) The ablation study of the components and hyperparameters in TrojanTO. (iii) The performance of persistent backdoor attacks. (iv) The impact of trigger perturbations. (v) The defense against TrojanTO.

### 6.1 ATTACK PERFORMANCE

We conducted extensive experiments to evaluate the performance of TrojanTO against established baselines, Baffle (Gong et al., 2024b) and IMC (Pang et al., 2020), across six diverse D4RL environments (Fu et al., 2020). The evaluation averaged results over three random seeds and three distinct target actions for three TO model variants. The aggregated performance metrics are presented

in Table 4. Overall, TrojanTO achieved an outstanding average CP of 0.701. This represents a substantial improvement of approximately 105.0% compared to Baffle (0.342 CP) and a significant 27.2% gain over IMC (0.551 CP).

Specifically, TrojanTO's efficacy is further highlighted by ASR and efficiency. It attained a high ASR of 0.719 while requiring a remarkably low average data poisoning rate of merely **0.3%**. In contrast, Baffle only reached an ASR of 0.369 despite a considerably higher 10% poisoning rate, underscoring TrojanTO's superior stealth and attack efficiency. Furthermore, TrojanTO also excels in maintaining BTP, achieving an average of 0.914. This is notably higher than Baffle (0.792) and IMC (0.853).

TrojanTO also exhibits consistent robustness and stability across varied tasks and TO model archi-tectures. Conversely, baseline methods demonstrate notable vulnerabilities in specific settings. For instance, when deployed with the DT model, the CP of IMC drastically reduces to a mere 0.013 in the `Hopp` and 0.133 in the `Ant`. Furthermore, Baffle exhibits a complete performance collapse in the `Walk` when against DT. Such critical shortcomings in baseline methods underscore their limited applicability and potential unreliability in TO models and highlight the superior reliability of TrojanTO. Complete results are detailed in Appendix K.3.

## 6.2 Ablation Study

To assess the contribution of each component within the TrojanTO, we conduct comprehensive ablation studies on the three modules illustrated in Figure 2. Specifically, TrojanTO w/o TF refers to the method that excludes trajectory filtering. TrojanTO w/o BP denotes the method removes batch poisoning, where the trigger is applied to all states within the poisoned batch. TrojanTO w/o AT denotes the method without alternate training, wherein the trigger learning is performed for an equal number of iterations before model updates.

Table 5: Average results of module ablation. Complete results can be seen in Table 23.

|  | TrojanTO w/o TF | | | TrojanTO w/o BP | | | TrojanTO w/o AT | | | **TrojanTO** | | |
|---|---|---|---|---|---|---|---|---|---|---|---|---|
|  | ASR | BTP | CP | ASR | BTP | CP | ASR | BTP | CP | ASR | BTP | CP |
| DT | 0.669 | 0.816 | 0.631 | 0.701 | 0.810 | 0.626 | 0.437 | **0.890** | 0.430 | **0.713** | 0.848 | **0.649** |
| DC | 0.623 | 0.860 | 0.597 | 0.312 | 0.848 | 0.363 | 0.470 | 0.904 | 0.484 | **0.631** | 0.947 | **0.640** |
| GDT | 0.742 | 0.873 | 0.742 | 0.571 | 0.851 | 0.562 | 0.614 | 0.940 | 0.637 | **0.814** | 0.948 | **0.814** |
| **Average** | 0.678 | 0.850 | 0.657 | 0.528 | 0.836 | 0.517 | 0.507 | 0.911 | 0.517 | **0.719** | **0.914** | **0.701** |

Table 5 presents the component-level ablation study results. Generally, all modules contribute positively. Specifically, for ASR, the 'BP' and 'AT' components exert a substantial influence: removing them causes ASR to decrease from 0.719 to 0.528 and 0.507 respectively. Regarding BTP, the 'TF' and 'BP' components are highly impactful, with their exclusion leading to a BTP reduction from 0.914 to 0.850 and 0.836 respectively. These findings confirm that the 'AT' component enhances attack effectiveness, while the 'TF' and 'BP' components, guided by the principle of precise poisoning, contribute significantly to attack stealth. We also conducted parameter-level ablation studies and investigated the impact of varying poisoning rates. These results are presented in Appendix J.

## 6.3 Persistent Backdoor Attack

A *persistent backdoor attack* is defined as an action-level backdoor where, once the trigger $\delta$ (applied to $s_{t-k}$) is activated, the target action will output consistently for the subsequent $k$ time steps. Unlike policy-level attacks, its malicious effect is persistent for a fixed duration of $k$ steps.

Table 6 demonstrates the efficacy of the persistent TrojanTO backdoor. The results confirm that upon trigger activation, the model consistently executes the

Table 6: The CP of the persistent backdoor attack with TrojanTO. The target type is '1'.

| k | Hopp | Half | Walk |
|---|---|---|---|
| 0 | $0.922_{\pm 0.000}$ | $0.972_{\pm 0.000}$ | $0.993_{\pm 0.000}$ |
| 5 | $0.898_{\pm 0.000}$ | $0.965_{\pm 0.000}$ | $0.876_{\pm 0.000}$ |
| 10 | $0.847_{\pm 0.012}$ | $0.954_{\pm 0.001}$ | $0.928_{\pm 0.000}$ |
| 15 | $0.880_{\pm 0.001}$ | $0.948_{\pm 0.000}$ | $0.973_{\pm 0.000}$ |

target action for the specified duration. Crucially, this sustained malicious behavior is maintained with only a minor degradation in CP as the persistence duration increases. However, the maximum

duration is fundamentally bounded by the TO model's finite context window (e.g., fewer than 20 steps). Beyond this context, the trigger is pushed out of context, causing the backdoor to deactivate.

### 6.4 TRIGGER PERTURBATIONS

To assess the robustness of the backdoor under environmental uncertainties, we inject multiplicative noise on the trigger, where each dimension is scaled by $(1 + \eta_d)$, where $\eta_d \sim \mathcal{U}(-\epsilon, \epsilon)$. $\mathcal{U}$ denotes a uniform distribution over the interval $(-\epsilon, \epsilon)$, and $\epsilon$ is the relative noise level.

The results shown in Table 7 reveal that the backdoor exhibits a gradual degradation in performance when subjected to perturbations, rather than an abrupt failure. This is consistent with the inherent smoothness of continuous models. The robustness to noise significantly amplifies the potential real-world security threats, as it allows adversaries to successfully activate the backdoor trigger even under diverse noisy conditions. However, as highlighted in (Guo et al., 2023), this robustness can also inadvertently lead to the emergence of pseudo triggers, which may ultimately compromise the stealthiness of the attack.

Table 7: ASR under trigger perturbations. The trigger dimensions are (1,2,3). The target type is '1'.

| $\eta_d$ | Hopp | Half | Walk |
|---|---|---|---|
| 0% | $0.895_{\pm 0.000}$ | $1.000_{\pm 0.000}$ | $0.980_{\pm 0.001}$ |
| 1% | $0.895_{\pm 0.000}$ | $1.000_{\pm 0.000}$ | $0.970_{\pm 0.001}$ |
| 5% | $0.885_{\pm 0.000}$ | $1.000_{\pm 0.000}$ | $0.897_{\pm 0.005}$ |
| 10% | $0.870_{\pm 0.000}$ | $1.000_{\pm 0.000}$ | $0.777_{\pm 0.025}$ |

### 6.5 DEFENSE

The paradigm shift from discrete to continuous action spaces introduces profound changes to the characteristics of backdoors against TO models. This motivated us to establish defenses against backdoor attacks in this new setting. We tested several baseline defense methods, including weight pruning, provable defense (Bharti et al., 2022), spectral analysis, activation clustering (Chen et al., 2018), and fine-tuning. Our results show that fine-tuning is the most effective defense, while the other tested methods proved largely ineffective in mitigating our TrojanTO attack. Detailed descriptions of these methods, experimental results, and ablation studies are provided in Appendix B.1.

## 7 CONCLUSION

This paper proposes TrojanTO, a novel post-training, action-level backdoor attack framework for TO models. It demonstrates effectiveness across different RL tasks, TO models, and attack scenarios. A comprehensive investigation reveals that the core of action-level backdoors against TO models lies in the design of triggers rather than in reward manipulation. Guided by this insight, TrojanTO leverages a consistency poisoning strategy to construct backdoors with a minimal attack budget, while maintaining a negligible impact on the agent's benign performance. We hope this study facilitates further research into the security of TO models and raises community awareness.

## 8 REPRODUCIBILITY STATEMENT

To ensure reproducibility, we have included the source code in the supplementary materials. All experimental details, including hyperparameter settings, dataset specifications, and implementation specifics for our proposed attack, are documented in Appendix C and I. Our code is available at `https://github.com/AndssY/TrojanTO`.

### ACKNOWLEDGMENT

This work is supported by the National Natural Science Foundation of China (Grant No. U25B2047, No. 72301289, No. 62576351, No. 62576364, No. 62441619, No. 62411540034, No. U2441239, and No. U24A20336), Shenzhen Basic Research Project (Natural Science Foundation) Basic Research Key Project (No. JCYJ20241202124430041), Shenzhen Science and Technology Program(NO.SYSRD20250529113401002).

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

# SUPPLEMENTARY MATERIAL FOR
# TROJANTO: ACTION-LEVEL BACKDOOR ATTACKS AGAINST TRAJECTORY OPTIMIZATION MODELS

# A    RELATED WORK

## A.1    POST-TRAINING BACKDOORS

While early backdoor research focused on training-time attacks, a more flexible and practical threat vector has emerged in the form of post-training attacks (Wu et al., 2023a; Meiklejohn et al., 2025). These attacks target pre-trained benign models and are highly efficient (Li et al., 2021; Hong et al., 2022; Guo et al., 2024a). An adversary only needs write access to the model weights and a minimal data budget to rapidly implant a backdoor. Their low cost and high speed make them a potent threat vector within the supply chain, especially in scenarios involving model sharing and third-party fine-tuning.

Despite its prevalence in other fields, this class of post-hoc vulnerability has been largely overlooked in RL. However, this oversight is becoming increasingly dangerous as large-scale models (e.g., TO models) become foundational for solving real-world decision-making tasks (Reed et al., 2022; Vuong et al., 2023), the community will inevitably rely on these pre-trained models. This makes it imperative to investigate security vulnerabilities in the post-training supply chain. The scenarios where a post-training attack could occur include: (1) An organization downloads a state-of-the-art pre-trained policy from a public platform like CleanRL (Huang et al., 2022) in Hugging Face. An adversary has previously compromised this model by fine-tuning it with a tiny, malicious dataset, embedding a hidden backdoor before it was uploaded. (2) A malicious employee with legitimate access to a company's model weights uses a small, crafted dataset to covertly perform post-training fine-tuning, embedding a backdoor for later exploitation. (3) An adversary first embeds a subtle backdoor via a during-training attack. Later, a post-training attack is used to implant a second backdoor into the victim model, creating a layered threat or adapting the model for a new malicious objective.

## A.2    MORE THREATS IN REINFORCEMENT LEARNING

In addition to backdoors, various other threats exist. For instance, adversarial perturbations can be introduced to the observations (Behzadan & Munir, 2017; Liu & Lai, 2023; Bai et al., 2024), actions (Lee et al., 2020), or rewards (Wu et al., 2023b; Xu et al., 2024) of the victim, thereby disrupting decision-making. Notably, reward poisoning can extend to safety alignment in RLHF (Baumgärtner et al., 2024; Pathmanathan et al., 2025), posing significant risks. Furthermore, adversarial policies (Gleave et al., 2020; Wang et al., 2023; Ma et al., 2024) can be employed to indirectly generate adversarial perturbations that disrupt the victim's decision-making. For instance, (Wang et al., 2023) successfully defeated the superhuman-level Go AI by using adversarial policies, highlighting the security threats these vulnerabilities pose to real-world decision-making systems.

There is also increasing attention on privacy protection concerning policies (Chen et al., 2021a), trajectories (Du et al., 2024; Gong et al., 2024a), and environments (Ye et al., 2025). It is essential to safeguard sensitive information in decision-making systems.

# B    DISCUSSION

## B.1    DEFENSES

This appendix details our evaluation of various backdoor defenses against the TrojanTO attack. We assessed five baseline methods: weight pruning, provable defense (Bharti et al., 2022), spectral analysis, activation clustering (Chen et al., 2018), and fine-tuning. For all experiments, we used the Decision Transformer (DT) as the victim model, a fixed random seed of '1' and a target action of '1'.

**Weight Pruning.** We evaluated two structured pruning methods. (1) Activation-based Pruning: Removes neurons whose activation changes most significantly between clean and triggered inputs. (2) Magnitude-based Pruning: Removes neurons with the lowest L1-norm weight magnitude.

As shown in Table 8, our experiments show that magnitude-based pruning is completely ineffective, failing to reduce the Attack Success Rate (ASR). Activation-based pruning slightly reduces ASR but at the cost of a severe drop in Benign Task Performance (BTP). Simple weight pruning is a poor trade-off, as it sacrifices significant BTP for incomplete and unreliable backdoor removal. This

outcome suggests that backdoors in continuous action spaces may be different from those in discrete domains like classification.

Table 8: ASR and BTP before and after applying defense at various pruning rates in Hopp, Walk, and Half. The best score after defense is highlighted in bold.

| Method | Rate | Before Defense | | After Defense | |
|---|---|---|---|---|---|
| | | ASR | BTP | ASR | BTP |
| activation | 0.01 | 1.000 | 0.847 | 0.920 | 0.674 |
| | 0.02 | 1.000 | 0.882 | 0.667 | **0.760** |
| | 0.03 | 1.000 | 0.846 | 0.667 | 0.584 |
| | 0.04 | 1.000 | 0.850 | 0.667 | 0.483 |
| | 0.05 | 1.000 | 0.628 | 0.573 | 0.407 |
| | 0.10 | 1.000 | 0.717 | 0.573 | 0.438 |
| | 0.15 | 1.000 | 0.687 | 0.440 | 0.210 |
| | 0.20 | 1.000 | 0.723 | **0.333** | 0.379 |
| magnitude | 0.01 | 1.000 | 0.836 | 1.000 | 0.663 |
| | 0.02 | 1.000 | 0.865 | 1.000 | **0.804** |
| | 0.03 | 1.000 | 0.870 | 1.000 | 0.579 |
| | 0.04 | 1.000 | 0.845 | 1.000 | 0.451 |
| | 0.05 | 1.000 | 0.782 | 1.000 | 0.503 |
| | 0.10 | 1.000 | 0.817 | 1.000 | 0.558 |
| | 0.15 | 1.000 | 0.807 | 1.000 | 0.344 |
| | 0.20 | 1.000 | 0.802 | 1.000 | 0.290 |

**Provable Defense.** This defense sanitizes the incoming state by projecting it onto a safe subspace. The subspace is built via SVD on a few clean states to identify their principal components, effectively learning the manifold of clean states. During deployment, this projection filters out-of-distribution triggers before the state reaches the model, aiming to neutralize the attack.

As shown in Table 9, the defense successfully reduces the ASR to near-zero. However, it also destroys the BTP, rendering the agent unusable for its intended task. This failure likely stems from the core assumption of the safe subspace not holding in continuous state spaces. Consequently, the projection discards not only the trigger but also critical state features, rendering the agent unusable.

Table 9: ASR and BTP before and after applying defense with different quantities of clean trajectories and projection dimensions in Hopp, Walk, and Half.

| # Clean data | Dimension | Before Defense | | After Defense | |
|---|---|---|---|---|---|
| | | ASR | BTP | ASR | BTP |
| 10 | 5 | 1.000 | 0.838 | 0.000 | 0.011 |
| | 8 | 1.000 | 0.882 | 0.000 | 0.050 |
| | 10 | 1.000 | 0.841 | 0.093 | 0.124 |
| 50 | 5 | 1.000 | 0.908 | 0.000 | 0.011 |
| | 8 | 1.000 | 0.825 | 0.000 | 0.039 |
| | 10 | 1.000 | 0.897 | 0.167 | 0.112 |
| 100 | 5 | 1.000 | 0.878 | 0.000 | 0.017 |
| | 8 | 1.000 | 0.834 | 0.000 | 0.083 |
| | 10 | 1.000 | 0.799 | 0.188 | 0.161 |
| all | 5 | 1.000 | 0.916 | 0.000 | 0.016 |
| | 8 | 1.000 | 0.857 | 0.000 | 0.010 |
| | 10 | 1.000 | 0.856 | 0.333 | 0.049 |

**Spectral Analysis.** We use spectral analysis on internal network activations to identify the "spectral signature" of normal, benign data. Inputs that cause activations to deviate significantly from this signature, such as those with a trigger, are then flagged as malicious.

As shown in Table 10, the spectral analysis detector is impractical due to the true positive rate (TPR) exhibiting significant variance across different environments and hyperparameters.

Table 10: TPR and FPR in Hopp, Half, and Walk at different detection thresholds.

| Env | threshold | TPR | FPR | Env | threshold | TPR | FPR | Env | threshold | TPR | FPR |
|-----|-----------|-----|-----|-----|-----------|-----|-----|-----|-----------|-----|-----|
| | 1.0 | 0.00 | 0.08 | | 1.0 | **1.00** | 0.37 | | 1.0 | 0.01 | 0.32 |
| | 1.5 | 0.00 | 0.05 | | 1.5 | **1.00** | 0.15 | | 1.5 | 0.00 | 0.12 |
| Hopp | 2.0 | 0.00 | 0.04 | Half | 2.0 | **1.00** | 0.03 | Walk | 2.0 | 0.00 | 0.04 |
| | 2.5 | 0.00 | 0.03 | | 2.5 | 0.00 | 0.00 | | 2.5 | 0.00 | 0.01 |
| | 3.0 | 0.00 | 0.02 | | 3.0 | 0.00 | 0.00 | | 3.0 | 0.00 | 0.00 |

**Activation Clustering.** Consistent with Baffle (Gong et al., 2024b), we employed Activation Clustering (AC) (Chen et al., 2018) as a detection-based defense. Remarkably, AC was entirely incapable of detecting the backdoor within TO models, resulting in a Precision and a Recall of 0%.

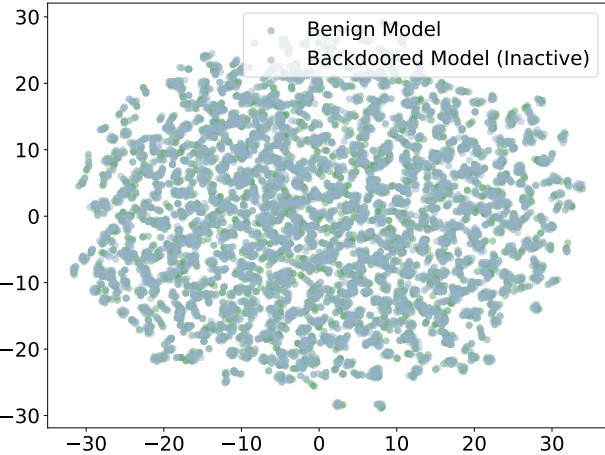

Figure 3: t-SNE result of the activations.

To elucidate this striking result, we performed a t-SNE visualization (Maaten & Hinton, 2008) of the internal activations produced by benign and backdoored models on clean trajectories. As illustrated in Figure 3, the t-SNE clusters from the two models are virtually indistinguishable. This observation explains why AC is unable to distinguish between a compromised model and a benign one.

We attribute this inseparability to a distinctive property of the TO model architecture. After the input tokens $(s, a, \hat{R})$ are embedded and concatenated, the variance of activations across different input types, which we term inter-type variance, substantially exceeds the intra-type variance induced by the trigger perturbation (i.e., between $s$ and $s + \delta$).

Consequently, the model's internal activation space is dominated by the variations between input modalities $(s, a, \hat{R})$, effectively masking the subtle changes introduced by the trigger. This inherent characteristic renders activation-based detection methods ineffective and exposes a fundamental security vulnerability in TO models.

Table 11: Fine-tuning Defense Performance in Half. Results show ASR and BTP before and after fine-tuning on 10 clean trajectories.

| Steps | 0 | 1000 | 2000 | 3000 | 4000 | 5000 | 6000 | 7000 | 8000 | 9000 | 10000 |
|-------|-----|------|------|------|------|------|------|------|------|------|-------|
| BTP | 0.964 | 0.986 | 0.971 | 0.985 | 0.948 | 0.950 | 0.920 | 0.978 | 0.892 | 0.928 | 0.917 |
| ASR | 1.000 | 1.000 | 1.000 | 1.000 | 1.000 | 1.000 | 0.960 | 0.760 | 0.480 | 0.280 | 0.040 |

**Fine-tuning.** Fine-tuning serves as an effective defense that suppresses the backdoor attack. As shown in Table 11, using only 10% of the original training steps, it reduces the ASR to near zero while maintaining the BTP. However, while it successfully reduces the ASR, a residual effect from the attack persists, as the output action's continued proximity to the target action.

## B.2 LIMITATION

This paper explores the efficient implantation of backdoors into trajectory optimization (TO) models in offline reinforcement learning (offline RL). We introduce TrojanTO, which enables precise manipulation of TO models under various target action conditions with just a small amount of data.

(1) However, it is still necessary to fine-tune the models for embedding the backdoor. Recently, a series of model editing methods (Bau et al., 2020; Meng et al., 2022; Wang et al., 2024) have been proposed. These model editing techniques can be applied to implant backdoors into Large Language Models (Guo et al., 2024a;b) and require no fine-tuning or extensive training; instead, they can inject backdoors with just a small number of samples and a few seconds of processing time. Applying model editing methods for backdoor implantation in RL models is a promising approach.

(2) We find that both the trigger dimensions and the selection of the target action significantly affect the backdoor's efficacy. However, there is currently a lack of automated methods for selecting trigger dimensions and evaluation methods for the difficulty of target actions. We explored several gradient-based methods for selecting the backdoor's dimensions. Nevertheless, there was no correlation between the dimension chosen based on gradient information and the backdoor's effectiveness. Future work could focus on addressing these two aspects.

(3) This paper explores the security threats posed by action-level backdoor attacks on TO models in continuous state spaces. Extending backdoor attacks to settings with image-based states and continuous actions is a crucial step towards evaluating their real-world implications, as it enables the study of triggers with greater physical plausibility.

## B.3 ETHICS STATEMENT

The primary goal of this work is to advance the security of machine learning systems. We acknowledge the dual-use nature of our proposed backdoor attack on Decision Transformers. While the technique could be exploited for malicious purposes, our core motivation is defensive. By demonstrating a concrete and potent vulnerability, we aim to raise awareness within the research community and provide a clear benchmark for the development and evaluation of effective countermeasures. We believe that proactively identifying and understanding such threats is an essential and responsible step toward building more robust and trustworthy AI systems.

## B.4 THE USE OF LARGE LANGUAGE MODELS

We utilized a Large Language Model to assist with grammar checking, spell checking, and improving the overall clarity and readability of this paper. Furthermore, we employed the model to generate auxiliary code snippets, such as statistical analysis and data visualization.

## C EXPERIMENTAL SETUP

## C.1 ENVIRONMENTS AND DATASETS

All datasets utilized in our experiments are sourced from the D4RL benchmark suite (Fu et al., 2020), which provides a standardized collection of datasets for offline RL research. These datasets cover a range of challenging tasks, including continuous control, navigation, and complex manipulation. Below, we detail the specific environments and datasets employed.

**MuJoCo Locomotion.** The MuJoCo (Todorov et al., 2012) locomotion tasks are standard benchmarks for evaluating RL algorithms in continuous control. Following the experimental setup of Baffle (Gong et al., 2024b), we selected the following D4RL datasets: `Hopper-Medium-Expert-v2`, `HalfCheetah-Medium-v2`, and `Walker2D-Medium-v2`. These datasets represent varying

qualities of data: 'Medium' datasets are generated by a SAC agent trained to a moderate level of performance, while 'Medium-Expert' datasets mix data from a medium-level policy with data from an expert policy. Hereafter, we refer to these environments and their corresponding datasets as `Hopp`, `Half`, and `Walk` respectively.

**AntMaze Navigation.**   The AntMaze environments present challenging navigation tasks characterized by sparse rewards, where an ant-like agent must traverse complex mazes. We focus on the `AntMaze-Umaze-v2` dataset (hereafter `Ant`). The Umaze designation within this dataset indicates tasks set in simpler, U-shaped maze configurations.

**Kitchen Manipulation.**   The Kitchen environment involves multi-stage, long-horizon manipulation tasks where a multi-jointed arm needs to interact with various objects in a kitchen setting (e.g., opening a microwave, turning on a burner). We used the dataset `Kitchen-Partial-v0`, which may only contain demonstrations for some of the sub-tasks. Hereafter, we refer to this environment and its corresponding dataset as `Kit`.

**Pen Manipulation.**   The Pen environment centers on dexterous manipulation, requiring a simulated anthropomorphic hand to reorient a pen to a target configuration. We employ the `Pen-Cloned-v1` dataset (hereafter `Pen`), which is generated by a policy trained to imitate a suboptimal demonstrator.

**Observation and Action Space.**   Finally, the specifications of the observation and action spaces for each environment are provided in Table 12. As is evident, both the observation and the action output for each environment are high-dimensional continuous vectors. A fundamental prerequisite for an effective action-level backdoor is that all components of the agent's outputted action must achieve proximity to the corresponding components of the target action. Consequently, the high-dimensional and continuous nature of these action spaces presents considerable challenges to the successful implementation of action-level backdoor attacks.

**Poisoning Rate.**   For the TrojanTO attack, the adversary utilizes 10 trajectories to implant the backdoor during the post-training phase. These trajectories can be obtained by interacting with the environment using the target TO model or another agent on the same task. Since only a small number of trajectories is needed, these interactions can be seen as routine testing. For instance, it is common practice to run a few test episodes after loading a model to verify its performance and ensure it has been loaded correctly. An adversary's data collection perfectly mimics this type of routine check, making it stealthy and practically undetectable.

In our experiments, we simulate this process by randomly sampling from the provided dataset. As shown in Table 12, the datasets contain an average of 3318 trajectories. Therefore, the poisoning rate for the TrojanTO attack is 10/3318, which is approximately 0.3%.

Table 12: Information for Each Environment.

| Information | Hopp | Half | Walk | Ant | Kit | Pen | Average |
|---|---|---|---|---|---|---|---|
| Observation Space | 11 | 17 | 17 | 27 | 59 | 45 | 29 |
| Action Space | 3 | 6 | 6 | 8 | 9 | 24 | 9 |
| # trajectories | 3213 | 1000 | 1190 | 10153 | 600 | 3754 | 3318 |

C.2   VICTIM MODEL AND HYPERPARAMETERS

**Victim Models.**   We select three prominent TO models as victims for our study: DT(Chen et al., 2021b), Graph Decision Transformer (GDT)(Hu et al., 2023), and Decision ConvFormer (DC)(Kim et al., 2023). DT serves as a foundational TO model. GDT enhances DT by explicitly modeling input sequences as causal graphs, thereby capturing potential dependencies between different concepts and facilitating the learning of temporal and causal relationships. DC, based on the MetaFormer(Yu et al., 2022) architecture, offers another distinct approach. DC addresses compatibility issues between attention modules and Markov Decision Processes (MDPs) by employing local convolution filtering as a token mixer, which effectively captures inherent local correlations within RL datasets. Both

GDT and DC represent significant advancements in TO model design. Furthermore, each model was trained using three different random seeds to account for training variability.

**Hyperparameters.** Table 13 details the hyperparameters for DT, GDT, and DC on the D4RL tasks. To ensure a fair and consistent comparison, our hyperparameter settings are largely aligned with those established for the original DT implementation. This includes shared parameters such as the number of transformer layers, the number of multi-head self-attention heads, embedding dimensions, as well as learning rate and optimizer configurations. For hyperparameters unique to GDT and DC, we adopted the default settings as reported in their respective original papers or official implementations.

Table 13: Hyperparameters of three TO models in D4RL tasks.

| Hyperparameters | Value |
|---|---|
| Layers | 3 |
| Embedding dimension | 128(DT, GDT)  256(DC) for MuJoCo tasks |
|  | 256(DT, GDT, DC) for other tasks |
| Batch size | 64 |
| Context length $K$ | 20 |
| Dropout | 0.1 |
| Learning rate | $10^{-4}$ |
| Grad norm clip | 0.25 |
| Weight decay | $10^{-4}$ |
| Learning rate decay | Linear warmup for first $10^5$ training steps |
| num_eval_episodes | 100 |
| max_iters | 10 |
| num_steps_per_iter | 10000 |

**Raw Performance.** The raw return scores for the DT, GDT, and DC models, averaged over three random seeds per environment, are detailed in Table 14. The benign task performance (BTP) is subsequently defined as the current reward normalized by its corresponding baseline, with the final quotient clipped to the unit interval [0, 1].

Table 14: Raw return scores of three TO models.

|  | Hopp | Half | Walk | Ant | Kit | Pen |
|---|---|---|---|---|---|---|
| DT | 3081 | 4994 | 3366 | 0.58 | 1.39 | 1908 |
| DC | 3054 | 4731 | 3001 | 0.55 | 0.46 | 2229 |
| GDT | 3358 | 4477 | 2851 | 0.55 | 0.58 | 1966 |

## C.3    TRAINING RESOURCES

We use the NVIDIA GeForce RTX 3090 and RTX 4090 to train each model. Models in each environment are trained three times with different seeds. During the backdoor implantation process, three distinct target actions were evaluated simultaneously to ensure the effectiveness of the backdoor attack method. To provide a comprehensive overview, the main results are presented in Table 4. In total, this extensive experimental setup involved conducting $3 \times 3 \times 6 + 3 \times 3 \times 3 \times 3 \times 6 = 540$ unique experiments.

## D    ALGORITHM

Algorithm 1 summarizes the implementation details of the TrojanTO method.

---

**Algorithm 1** TrojanTO

---

**input** Pre-trained TO model $\pi$, target action $a^\dagger$, initial trigger $\delta$,
      trigger bounds $\delta_{\min}, \delta_{\max}$, max iterations $M$, inner iterations
      $N_1, N_2$,
      trajectory dataset $\{\tau_i\}_{i \in N}$, parameters $\mu, \alpha$.
  1: Select the trigger dimensions $M$ for optimization.
  2: // Initialization
  3: $\delta^k = Mask\_Initialize(M), \tilde{\pi}^k = \pi$.
  4: $F_\tau = Trajectory\_Filtering(\{\tau_i\}_{i \in N})$
  5: $B_c, B_p = Batch\_Poisoning(F_\tau)$
  6: **for** $k \leq M//2$ **do**
  7:     **for** $i \leq N_1$ **do**
  8:        $\delta_i^k = Trigger\_Learning(\delta_{i-1}^k, \tilde{\pi}^k, \mu, \alpha, B_p, \delta_{\min}, \delta_{\max})$.
  9:     **end for**
10:     **for** $j \leq N_2$ **do**
11:        $\tilde{\pi}_j^k = Parameter\_Updating(\tilde{\pi}_{j-1}^k, \delta^k, B_c, B_p)$.
12:     **end for**
13: **end for**
14: **for** $j \leq M//2 * N_2$ **do**
15:     $\tilde{\pi}_j^k = Parameter\_Updating(\tilde{\pi}_{j-1}^k, \delta^k, B_c, B_p)$.
16: **end for**
**output** $\delta^k$ and $\tilde{\pi}^k$.

---

# E  TWO TYPES OF BACKDOOR ATTACKS

**Policy-Level Backdoor.** This type of backdoor targets the long-term objectives of the victim agent, where each trigger is linked to a specific target policy.

For instance, an adversary could activate the backdoor at a critical moment to redirect an autonomous vehicle's destination from a school to a hospital, regardless of the route taken.

**Action-Level Backdoor.** This type of backdoor concentrates on precise control over individual actions at each time step, with each trigger corresponding to a desired action. Action-level backdoor attacks offer a versatile range of malicious capabilities. First, they can cause catastrophic failure by compromising just a single critical action. Second, through repeated trigger activations, they allow an adversary to orchestrate complex, multi-step malicious manipulations. Finally, they provide significant tactical flexibility, as different long-term objectives can be pursued simply by altering the activation patterns of the trigger.

For example, consider an adversary targeting an autonomous driving agent. The backdoor's capabilities can be deployed at three distinct levels of sophistication:

(1) The adversary could compel the agent to make an abrupt, dangerous turn at a pivotal moment, such as a crowded intersection. This single malicious action is designed to directly cause a catastrophic outcome like a collision.

(2) The adversary could execute a multi-step plan to subtly guide the vehicle off its intended route to a malicious location. This might involve a sequence of seemingly innocuous actions: ignoring the correct highway exit, taking an unusual side street, and finally stopping in a secluded area—a plan composed of multiple triggered actions.

(3) The adversary could use the same backdoor for an entirely different purpose. The adversary could dynamically alter the trigger activation strategy to redirect the vehicle to a totally different destination, all without retraining or re-implanting the backdoor. This demonstrates the ability to change the long-term malicious objective at will.

To substantiate that action-level backdoors are the foundational building blocks for complex policy manipulation, and to simultaneously assess the long-term effectiveness of TrojanTO, we conducted an in-depth investigation of various trigger activation strategies, adopting the methodology of Baffle.

Table 15: The performance of the backdoored model under various trigger activation strategies. Results are averaged over 5 random seeds.

| Method | BTP | Trigger Interval | | | Trigger Length | | |
|---|---|---|---|---|---|---|---|
| | | 10 | 20 | 50 | 5 | 10 | 20 |
| Baffle | 0.751 | $0.479_{\pm 0.014}$ | $0.558_{\pm 0.022}$ | $0.619_{\pm 0.011}$ | $0.514_{\pm 0.008}$ | $0.558_{\pm 0.014}$ | $0.484_{\pm 0.004}$ |
| TrojanTO | 0.866 | $0.162_{\pm 0.000}$ | $0.273_{\pm 0.002}$ | $0.398_{\pm 0.032}$ | $0.766_{\pm 0.039}$ | $0.018_{\pm 0.000}$ | $0.018_{\pm 0.000}$ |

The results in Table 15 demonstrate TrojanTO's ability to manipulate long-term rewards. While the backdoor by Baffle impact is moderate, the backdoor by TrojanTO causes a catastrophic performance collapse. Sustained triggers drive the agent's reward to nearly zero, proving that our action-level attack can successfully hijack the entire policy trajectory.

## F    THE IMPORTANCE OF TRIGGER DIMENSIONS

In tasks like image classification, models typically focus on different regions of the input image. A widely used approach to visualize this attention is through saliency maps. One such method, Grad-Cam, visualizes the model's focus by calculating the gradient of the predicted class score $y^c$ with respect to the feature map $A^k$ produced by the last convolutional layer. The gradient with respect to $A^k$ is computed as follows:

$$\frac{\partial y^c}{\partial A^k}.$$

Subsequently, the gradients are aggregated by averaging over all spatial locations in the feature map:

$$\alpha^k = \frac{1}{Z} \sum_{i,j} \frac{\partial y^c}{\partial A_{ij}^k},$$

where $Z$ is the total number of spatial positions in the feature map, and $\frac{\partial y^c}{\partial A_{ij}^k}$ represents the gradient of the class score with respect to the $(i, j)$-th spatial location in the feature map $A^k$.

The aggregated gradients $\alpha^k$ are then used as weights to combine the feature maps, which are summed across all channels:

$$L_{\text{GradCam}} = \text{ReLU}\left( \sum_k \alpha^k A^k \right).$$

The ReLU activation ensures that only the positively contributing regions are visualized, as negative influences are generally not informative for the interpretation of attention maps.

Finally, the resulting map $L_{\text{GradCam}}$ is upsampled to match the dimensions of the input image, generating a heatmap that highlights the areas of the image the model attends to when making predictions.

Inspired by this approach, we applied the Grad-Cam method to DT, and the resulting attention map is shown in Figures 4-6. The horizontal axis represents each decision time step, while the vertical axis represents the current time step. The model's attention to different input dimensions is depicted, which corresponds to the average gradient values computed using Grad-Cam. The DT model is an in-context model, and here we only visualize the model's attention to the current state, disregarding historical information.

In order to select trigger dimensions for the effective implantation of backdoors, we employed the Gradient-weighted Class Activation Mapping (Grad-CAM) method (Selvaraju et al., 2017) to obtain the heat map for different input dimensions of each time step.

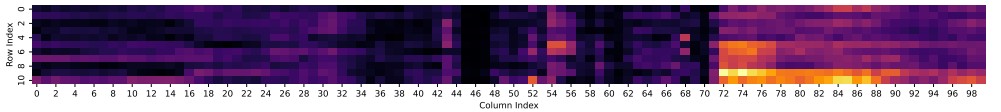

Figure 4: The heatmap of the Hopper environment, decision time steps from 20 to 120.

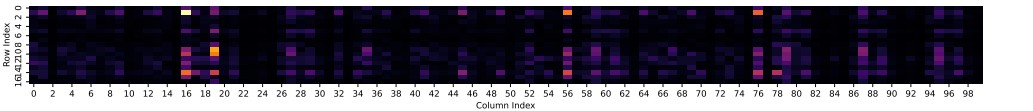

Figure 5: The heatmap of the Halfcheetah environment, decision time steps from 20 to 120.

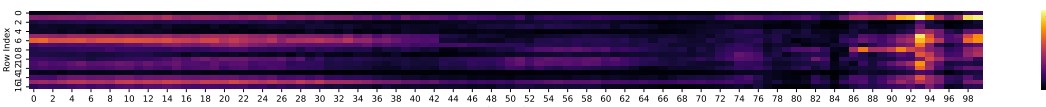

Figure 6: The heatmap of the Walker2d environment, decision time steps from 20 to 120.

It can be observed that the model's attention varies across different input dimensions at the same time step and shifts across different time steps. However, overall, certain dimensions consistently receive more attention from the model.

However, using these attention-focused dimensions as trigger dimensions does not improve the effectiveness of backdoor implantation. We identified the three least significant dimensions in the Walker2D environment: [13, 0, 4], and the three most significant dimensions: [1, 15, 6], while also randomly selecting some trigger dimensions. We observe that there is a significant variation in the ASR (Attack Success Rate) for different trigger dimensions, highlighting the importance of selecting the right trigger dimensions. However, regardless of whether the dimensions are highly significant or not, the backdoor implantation performance is inferior to that achieved with randomly chosen trigger dimensions. This indicates that further research is needed to select trigger dimensions for efficient backdoor implantation.

In addition, we also referred to the method that uses the Spectrum of the Neural Tangent Kernel (NTK) to predict fine-tuning performance (Afzal et al., 2024), which suggests a negative correlation between the Spectrum of NTK and fine-tuning performance. We computed the Spectrum of NTK under different triggers but found no significant correlation. Therefore, it is concluded that gradient information may not be effective for selecting the trigger dimensions. Future research could explore the use of search algorithms to efficiently identify the trigger dimensions.

## G   THE TRIGGERS' VALUES

We present the trigger values obtained using different methods for the dimensions (8, 9, 10). The 'Handcrafted Trigger' is designed through empirical experience. The 'Baffle Trigger' refers to the trigger values used in (Gong et al., 2024b). The 'Dataset Trigger' is derived by searching for the state values corresponding to the actions closest to the target action (arithmetic) in the dataset. The 'Learnable Trigger' is optimized using the MI-FGSM method, and the 'Over-bound Trigger' is generated by directly setting values that exceed the normal state space.

## H   THE TARGET TYPES

As shown in Table 17, we consider the following six target actions. These include boundary values ('1', '-1'), intermediate values ('0', '0.5staggered'), regular patterns ('arithmetic'), and randomly generated actions ('fixed random'). We selected three target actions, namely '1', 'arithmetic', and 'fixed random', to assess the effectiveness of the backdoor. However, future research could focus on developing a rapid method to evaluate the difficulty of various target actions.

Table 16: Trigger values used for different environments in trigger value's experience within the dimensions of (8,9,10).

|  | Halfcheetah | Walker2d |
| --- | --- | --- |
| Handcrafted Trigger | (0.479, 3.127, 4.433) | (-4.111, 2.209, -0.837) |
| Baffle Trigger | (4.561, -0.060, -0.114) | (2.022, -0.210, -0.374) |
| Dataset Trigger | (1.953, -0.264, -1.665) | (1.536,-0.033,2.571) |
| Learnable Trigger | ( -5.650, 6.800, -8.510) | (-3.750, -9.800, -9.800) |
| Over-bound Trigger | (-100, 100, 100) | (-100, 100, 100) |

Table 17: The type and corresponding value of the target action. The listed values are truncated to align with the action space dimension of each specific environment.

| Target Type | Action Value |
| --- | --- |
| '0' | [0, 0, 0, . . . , 0, 0, 0] |
| '1' | [1, 1, 1, . . . , 1, 1, 1] |
| '-1' | [-1, -1, -1, . . . , -1, -1, -1] |
| 'fixed random' | [0.497, 0.695, -0.711, -0.336, 0.141, -0.374, -0.450, 0.640, 0.160, -0.370, 0.177, 0.681, -0.625, -0.435, 0.714, -0.392, 0.944, -0.649, 0.266, -0.691, -0.167, 0.468, -0.390, 0.202] |
| 'arithmetic' | [0.0, 0.1, 0.2, 0.3, 0.4, 0.5, . . . ] |
| '0.5staggered' | [0.5, -0.5, 0.5, -0.5, 0.5, -0.5, . . . ] |

## I  THE IMPLEMENTATION DETAILS

This section provides a detailed setup for each experiment.

### I.1  TARGET ACTION'S IMPLEMENTATION DETAILS

For the investigation of the target action experiments. We employed the baffle attack (Gong et al., 2024b) for the Halfcheetah and Walker2d environments. The trigger dimensions selected are (8, 9, 10), with the trigger values specified in Table 16, and the action values corresponding to different target actions are also listed in Table 17. We modify 10% of the trajectories in the dataset by adding the trigger to the states, changing the actions to the target actions, and adjusting the reward value to 4. These backdoored data are then used for training. The training process consists of 20,000 steps, with evaluations conducted every 1,000 steps over 100 episodes. For the Hopper environment, we use TrojanTO. In the trigger learning process, the learning rate was set to 0.001, with a momentum of 0.9. The number of internal iterations was specified as 10, while the number of external iterations was set to 200. Model updates and trigger learning were alternated every 1,000 steps, and after 10,000 steps, only model updates were performed.

### I.2  TRIGGER DIMENSION'S IMPLEMENTATION DETAILS

We utilized TrojanTO for the implementation of the triggers. For the Halfcheetah environment, the learning rate during the trigger learning phase was set to 0.001, accompanied by a momentum of 0.9. The number of internal iterations was specified as 10, while the number of external iterations was set to 200. In the case of the Walker2d environment, the number of internal iterations was increased to 20, with external iterations set to 300.

### I.3  TRIGGER VALUE'S IMPLEMENTATION DETAILS

Since both the dimensions and values of the triggers have been predetermined, we employ TrojanTO without Trigger Learning (TrojanTO w/o TL), which means that we refrain from using trigger learning and alternate optimization. Instead, we utilize trajectory filtering and batch poisoning methods in TrojanTO.

### I.4  REWARD HACKING'S IMPLEMENTATION DETAILS

We use TrojanTO w/o BP. While maintaining the core architecture of TrojanTO w/o BP, we implement two critical modifications in reward hacking experiments: (1) During poisoned data construction, we systematically replace original reward values with predetermined experimental targets to create controlled reward manipulation scenarios; (2) We adopt a persistent alternating training protocol where trigger learning and parameter updating occur synchronously every epoch. All other hyperparameters are maintained identical to the original TrojanTO configuration to ensure experimental comparability.

## J  ABLATION STUDIES

### J.1  THE FORMULATION OF BATCH POISONING

The batch poisoning (BP) module of TrojanTO employs a joint optimization strategy, where the final loss is a weighted combination of the backdoor loss (Equation 5) and the clean loss (Equation 6). We performed an ablation study against a naive single-objective approach, which optimizes only Equation 6 across all data, including poisoned ones. As demonstrated in Table 18, our joint optimization method yields substantially higher ASR and CP.

Table 18: The comparison of different optimization formulations in the BP module. Experiments were performed on the `Hopp`, `Half`, `Walk` with three target action types: '-1', '1', and 'fixed-random'.

| Formulation | ASR | BTP | CP |
|---|---|---|---|
| Single-objective Optimization | 0.474 | **0.938** | 0.494 |
| Joint Optimization | **0.667** | 0.916 | **0.649** |

This joint optimization is crucial because it explicitly decouples the learning of benign and malicious behaviors. It instills a precise conditional policy by training the model to associate the trigger with the target action (Equation 5), while simultaneously ensuring that the agent's behavior remains unchanged on clean data (Equation 6). Conversely, the naive single-objective approach would face an ambiguous learning signal, leading to ineffective backdoor learning if only optimizing (Equation 6) over all data points (including poisoned ones).

### J.2  THE FREQUENCY OF ALTERNATING

We conducted ablation studies on the alternating training (AT) module, with results shown in the Table 19. The empirical analysis reveals that performance first increases and then decreases as the alternating frequency increases. Given the high volatility of RL tasks, we suggest reducing the alternating frequency when facing difficult tasks.

Table 19: Impact of Alternation Frequency on Performance. Results are averaged across three target actions ('1', 'arithmetic', 'fixed random'), three victim models (DT, DC, GDT), and three tasks (`Hopp`, `Half`, `Walk`). "Low Frequency" refers to 2 alternations between trigger learning and parameter updates, with others being similar.

| | ASR | BTP | CP |
|---|---|---|---|
| TrojanTO w/o AT | 0.59 | 0.93 | 0.59 |
| Low Frequency (2 alternation) | 0.78 | **0.94** | 0.76 |
| Middle Frequency (10 alternation) | **0.80** | **0.94** | **0.80** |
| High Frequency (100 alternation) | 0.66 | 0.90 | 0.66 |
| Highest Frequency (10000 alternations) | 0.53 | 0.82 | 0.51 |

## J.3 THE SELECTION OF BACKDOOR DATA

Intuitively, high-quality trajectories are crucial for training TO models, especially when the training data are limited. When failed trajectories are chosen for backdoor training, the model is likely to overfit these trajectories, leading to a significant degradation in performance on the normal task.

We conduct ablation experiments employing different data selection methods to select the trajectories used in backdoor training. As shown in Table 20, the 'Bad' method exhibits poor training data quality, resulting in significantly lower CP across most environments. In contrast, the 'Random' method demonstrates better performance. The 'Filtering' method achieves the highest CP, which demonstrates that high-quality trajectories are crucial for effective backdoor implantation.

Table 20: The CP under different data selection methods. The target type is '1'. The 'Bad' method retains only the shortest trajectories. The 'Random' method does not apply any trajectory filtering. The 'Filtering' method utilizes the previously mentioned trajectory filtering approach.

| Methods | Hopp | Half | Walk |
|---|---|---|---|
| Bad | $0.130_{\pm 0.000}$ | $0.970_{\pm 0.000}$ | $0.401_{\pm 0.007}$ |
| Random | $0.344_{\pm 0.093}$ | $0.969_{\pm 0.000}$ | $0.943_{\pm 0.000}$ |
| Filtering | $\mathbf{0.670}_{\pm 0.127}$ | $\mathbf{0.972}_{\pm 0.000}$ | $\mathbf{0.993}_{\pm 0.000}$ |

## J.4 THE ATTACK BUDGET OF BAFFLE

We employ batch poisoning rather than trajectory poisoning to implant a backdoor efficiently. The batch poisoning method ensures consistent triggering of the backdoor during both the training and evaluation phases, thereby enhancing the efficiency and effectiveness of the poisoning process.

Table 21: The performance of the trajectory poisoning method with different poisoning rates and TrojanTO (ASR↑/ BTP↑/ CP↑).

| | | Trajectory Poisoning | | | | | TrojanTO |
|---|---|---|---|---|---|---|---|
| | Poisoning Rate | 10% | 30% | 50% | 70% | 90% | |
| Hopp | ASR | $0.623_{\pm 0.045}$ | $0.907_{\pm 0.002}$ | $0.960_{\pm 0.003}$ | $0.967_{\pm 0.002}$ | $\mathbf{1.000}_{\pm 0.000}$ | $0.860_{\pm 0.001}$ |
| | BTP | $0.986_{\pm 0.000}$ | $0.955_{\pm 0.000}$ | $0.972_{\pm 0.000}$ | $0.890_{\pm 0.003}$ | $0.848_{\pm 0.010}$ | $\mathbf{1.000}_{\pm 0.000}$ |
| | CP | $0.743_{\pm 0.031}$ | $0.930_{\pm 0.001}$ | $\mathbf{0.965}_{\pm 0.001}$ | $0.927_{\pm 0.003}$ | $0.914_{\pm 0.004}$ | $0.926_{\pm 0.000}$ |
| Half | ASR | $0.093_{\pm 0.002}$ | $0.203_{\pm 0.016}$ | $0.227_{\pm 0.027}$ | $0.300_{\pm 0.032}$ | $0.177_{\pm 0.035}$ | $\mathbf{1.000}_{\pm 0.000}$ |
| | BTP | $0.930_{\pm 0.000}$ | $0.938_{\pm 0.000}$ | $\mathbf{0.945}_{\pm 0.000}$ | $0.925_{\pm 0.000}$ | $0.892_{\pm 0.002}$ | $\mathbf{0.945}_{\pm 0.000}$ |
| | CP | $0.166_{\pm 0.005}$ | $0.316_{\pm 0.025}$ | $0.337_{\pm 0.040}$ | $0.418_{\pm 0.053}$ | $0.249_{\pm 0.055}$ | $\mathbf{0.972}_{\pm 0.000}$ |
| Walk | ASR | $0.000_{\pm 0.000}$ | $0.017_{\pm 0.000}$ | $0.030_{\pm 0.001}$ | $0.267_{\pm 0.035}$ | $\mathbf{1.000}_{\pm 0.000}$ | $\mathbf{1.000}_{\pm 0.000}$ |
| | BTP | $0.983_{\pm 0.000}$ | $0.970_{\pm 0.000}$ | $0.933_{\pm 0.002}$ | $0.542_{\pm 0.103}$ | $0.064_{\pm 0.000}$ | $\mathbf{0.987}_{\pm 0.000}$ |
| | CP | $0.000_{\pm 0.000}$ | $0.032_{\pm 0.001}$ | $0.056_{\pm 0.004}$ | $0.241_{\pm 0.031}$ | $0.121_{\pm 0.000}$ | $\mathbf{0.993}_{\pm 0.000}$ |

To validate the effectiveness of batch poisoning, we compare TrojanTO and trajectory poisoning methods with different poisoning rates. As shown in Table 21, the trajectory poisoning method's CP exhibits a pattern of initially increasing and then subsequently decreasing as the poisoning rate rises. This phenomenon can be primarily attributed to an increase in the ASR, while the BTP declines. Trajectory poisoning has not been successful across all environments. In contrast, TrojanTO demonstrates consistently robust performance across all three evaluated environments.

## J.5 THE ATTACK BUDGET OF TROJANTO

We performed an ablation study with varying attack budgets. As shown in Table 22, the CP values for attack budgets of 0.31%, 1.1%, 3.1%, and 16% are 0.63, 0.65, 0.70, and 0.70, respectively. An increased attack budget improves performance. However, beyond a certain point, it plateaus.

Table 22: The impact of the attack budget on TrojanTO attack (ASR↑ / BTP↑ / CP↑). The results are averaged across three seeds and two victim models (DT, DC). The experiments were conducted in the `Hopp` environment.

| | Target Action | | | |
| --- | --- | --- | --- | --- |
| Attack Budget | 1 | arithmetic | fixed random | Average |
| 0.31% (10 trajectories) | 1.00/0.89/0.94 | 0.48/0.91/0.47 | 0.47/0.82/0.48 | 0.65/0.87/0.63 |
| 1.1% (36 trajectories) | 1.00/0.92/0.95 | 0.41/0.93/0.50 | 0.48/1.00/0.49 | 0.63/0.95/0.65 |
| 3.1% (100 trajectories) | 0.96/0.98/0.98 | 0.56/1.00/0.63 | 0.49/1.00/0.50 | 0.67/0.99/**0.70** |
| 16% (500 trajectories) | 1.00/1.00/1.00 | 0.50/1.00/0.50 | 0.54/1.00/0.60 | **0.68**/1.00/**0.70** |
| 31% (1000 trajectories) | 0.94/1.00/0.97 | 0.32/1.00/0.37 | 0.58/1.00/0.67 | 0.61/**1.00**/0.67 |

## K    MORE RESULTS

### K.1    ASR AND BTP COMPARISON IN HOPPER

As shown in Figure 7, both ASR and BTP exhibit consistent trends throughout training, remaining largely unaffected by the variations in the manipulated reward signal. Consequently, the insensitivity to reward manipulation confirms its limit for backdooring TO models.

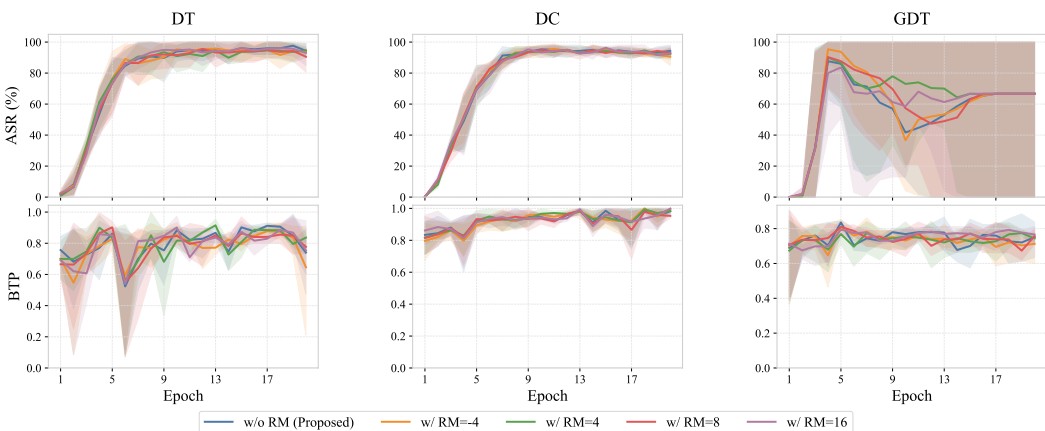

Figure 7: ASR and BTP comparison under different reward manipulation strategies across different TO models in Hopp. The trigger dimension is (8,9,10). The target type is '1'.

### K.2    COMPLETE ABLATION RESULT

To validate the individual contributions of TrojanTO's key components, we conducted a comprehensive ablation study. The results, averaged across three seeds and detailed in Table 23, demonstrate the integral role of each component in achieving the overall efficacy of TrojanTO.

Removing the module responsible for trigger generation (TrojanTO w/o TF) led to a notable decrease in average CP from 0.701 to 0.657, and ASR from 0.719 to 0.678. This indicates the significance of the specialized trigger design for effective attack execution. The absence of the adversarial training component (TrojanTO w/o AT) resulted in a more pronounced degradation, particularly in attack efficacy. Average ASR dropped sharply to 0.507, and CP reduced to 0.517, underscoring this module's critical contribution to enhancing the potency and stealth of the embedded backdoor. Similarly, omitting the benign preservation component (TrojanTO w/o BP) also significantly impacted overall performance. Average CP fell to 0.517, and ASR to 0.528. Crucially, this ablation also led to a marked decrease in BTP from 0.914 to 0.836, highlighting this module's dual function in not only supporting the attack's objectives but also in safeguarding the agent's original task capabilities.

These findings collectively affirm that each ablated module plays an indispensable and synergistic role in achieving the superior attack performance and robust benign task preservation characteristic of the complete TrojanTO framework.

Table 23: Complete results of module ablation study averaged over three seeds.

| Model | Env | TrojanTO w/o TF | | | TrojanTO w/o AT | | | TrojanTO w/o BP | | | **TrojanTO** | | |
|---|---|---|---|---|---|---|---|---|---|---|---|---|---|
| | | ASR | BTP | CP | ASR | BTP | CP | ASR | BTP | CP | ASR | BTP | CP |
| DT | Hopp | 0.019 | 0.645 | 0.030 | 0.054 | 0.899 | 0.081 | 0.508 | 0.769 | **0.374** | 0.362 | 0.882 | 0.365 |
| | Half | 0.998 | 0.953 | 0.975 | 1.000 | 0.953 | 0.976 | 0.963 | 0.958 | 0.958 | 1.000 | 0.982 | **0.991** |
| | Walk | 0.712 | 0.892 | 0.690 | 0.340 | 0.923 | 0.330 | 0.753 | 0.876 | 0.833 | 0.990 | 0.926 | **0.957** |
| | Ant | 0.657 | 0.768 | **0.588** | 0.282 | 0.822 | 0.270 | 0.349 | 0.860 | 0.350 | 0.296 | 0.843 | 0.302 |
| | Kit | 0.960 | 0.850 | **0.898** | 0.570 | 0.748 | 0.528 | 0.964 | 0.413 | 0.574 | 0.969 | 0.455 | 0.614 |
| | Pen | 0.667 | 0.787 | 0.605 | 0.374 | 0.995 | 0.392 | 0.667 | 0.987 | **0.667** | 0.661 | 1.000 | 0.664 |
| GDT | Hopp | 0.222 | 0.755 | 0.196 | 0.503 | 0.802 | 0.455 | 0.410 | 0.775 | 0.477 | 0.508 | 0.766 | **0.503** |
| | Half | 0.999 | 0.999 | **0.999** | 0.812 | 1.000 | 0.822 | 0.455 | 1.000 | 0.533 | 0.967 | 1.000 | 0.981 |
| | Walk | 0.441 | 1.000 | 0.482 | 0.340 | 0.981 | 0.346 | 0.358 | 0.950 | 0.396 | 0.418 | 1.089 | **0.486** |
| | Ant | 0.751 | 0.837 | **0.662** | 0.062 | 0.842 | 0.080 | 0.281 | 0.990 | 0.302 | 0.334 | 0.963 | 0.336 |
| | Kit | 0.683 | 0.749 | 0.658 | 0.564 | 0.883 | 0.638 | 0.166 | 0.435 | 0.185 | 0.889 | 0.887 | **0.881** |
| | Pen | 0.641 | 0.822 | 0.587 | 0.540 | 0.915 | 0.562 | 0.201 | 0.939 | 0.286 | 0.667 | 0.976 | **0.653** |
| DC | Hopp | 0.409 | 0.706 | 0.493 | 0.408 | 0.880 | 0.478 | 0.737 | 0.734 | 0.712 | 0.931 | 0.854 | **0.889** |
| | Half | 1.000 | 1.000 | **1.000** | 0.999 | 0.999 | 0.999 | 1.000 | 1.000 | **1.000** | 1.000 | 1.000 | **1.000** |
| | Walk | 0.861 | 0.960 | 0.859 | 0.824 | 0.983 | 0.861 | 0.723 | 0.797 | 0.631 | 0.995 | 0.982 | **0.988** |
| | Ant | 0.722 | 0.830 | **0.654** | 0.124 | 0.844 | 0.157 | 0.268 | 0.803 | 0.300 | 0.572 | 0.884 | 0.559 |
| | Kit | 1.000 | 1.000 | **1.000** | 0.983 | 0.960 | 0.970 | 0.357 | 0.815 | 0.378 | 0.960 | 0.982 | 0.969 |
| | Pen | 0.460 | 0.741 | 0.445 | 0.346 | 0.975 | 0.357 | 0.342 | 0.955 | 0.349 | 0.428 | 0.984 | **0.477** |
| **Average** | | 0.678 | 0.850 | 0.657 | 0.507 | 0.911 | 0.517 | 0.528 | 0.836 | 0.517 | **0.719** | **0.914** | **0.701** |

## K.3    COMPLETE PERFORMANCE OF TROJANTO AND BASELINES

To further scrutinize the robustness of TrojanTO, Table 24 details performance across three distinct target action types. TrojanTO consistently achieved the highest average CP across all three target action categories: 0.942 for target action '1', 0.536 for 'arithmetic', and 0.624 for 'fixed random'. These scores significantly surpass those of Baffle (0.745, 0.123, and 0.159, respectively) and IMC (0.759, 0.352, and 0.542, respectively). While the 'arithmetic' target action presented a greater challenge for all methods, as evidenced by generally lower CP values, TrojanTO maintained a substantial performance advantage over the baselines. This underscores TrojanTO's superior capability in effectively executing diverse and potentially complex malicious objectives, while also preserving commendable BTP across these varied attack strategies (average BTPs of 0.929, 0.912, and 0.901 for the respective target actions).

Table 24: The performance of TrojanTO and baselines with the different target action (ASR↑/ BTP↑/ CP↑). The results are averaged across three random seeds.

| target action '1' | | Baffle | | | IMC | | | TrojanTO | | |
|---|---|---|---|---|---|---|---|---|---|---|
| Model | Env | ASR | BTP | CP | ASR | BTP | CP | ASR | BTP | CP |
| DT | Hopp | 0.986 | 0.623 | 0.743 | 0.466 | 0.000 | 0.000 | 1.000 | 0.878 | 0.935 |
| | Half | 0.930 | 0.093 | 0.166 | 0.918 | 0.577 | 0.705 | 1.000 | 0.984 | 0.992 |
| | Walk | 0.983 | 0.000 | 0.000 | 0.508 | 0.000 | 0.000 | 1.000 | 0.973 | 0.986 |
| | Ant | 0.493 | 0.823 | 0.617 | 0.297 | 0.760 | 0.398 | 0.640 | 0.876 | 0.658 |
| | Kit | 1.000 | 0.659 | 0.788 | 0.963 | 0.786 | 0.859 | 0.997 | 0.488 | 0.654 |
| | Pen | 0.937 | 1.000 | 0.967 | 1.000 | 1.000 | 1.000 | 1.000 | 1.000 | 1.000 |
| GDT | Hopp | 0.987 | 0.771 | 0.861 | 0.990 | 0.832 | 0.904 | 1.000 | 0.831 | 0.908 |
| | Half | 0.600 | 0.939 | 0.727 | 1.000 | 1.000 | 1.000 | 1.000 | 1.000 | 1.000 |
| | Walk | 0.660 | 0.949 | 0.765 | 1.000 | 1.000 | 1.000 | 1.000 | 1.000 | 1.000 |
| | Ant | 0.920 | 0.998 | 0.955 | 0.503 | 0.882 | 0.565 | 1.000 | 1.000 | 1.000 |
| | Kit | 0.950 | 0.862 | 0.889 | 1.000 | 1.000 | 1.000 | 0.990 | 0.931 | 0.958 |
| | Pen | 0.996 | 1.000 | 0.998 | 1.000 | 1.000 | 1.000 | 1.000 | 1.000 | 1.000 |
| DC | Hopp | 0.762 | 0.947 | 0.844 | 0.762 | 0.947 | 0.844 | 1.000 | 0.885 | 0.938 |
| | Half | 0.835 | 0.620 | 0.711 | 0.835 | 0.620 | 0.711 | 1.000 | 1.000 | 1.000 |
| | Walk | 0.838 | 0.583 | 0.686 | 0.838 | 0.583 | 0.686 | 1.000 | 0.996 | 0.998 |
| | Ant | 0.750 | 0.768 | 0.759 | 0.993 | 1.000 | 0.997 | 0.997 | 0.909 | 0.950 |
| | Kit | 1.000 | 1.000 | 1.000 | 1.000 | 1.000 | 1.000 | 0.993 | 0.974 | 0.983 |
| | Pen | 0.890 | 0.998 | 0.940 | 1.000 | 1.000 | 1.000 | 1.000 | 1.000 | 1.000 |
| **Average** | | 0.862 | 0.757 | 0.745 | 0.837 | 0.777 | 0.759 | **0.979** | **0.929** | **0.942** |
| target action 'arithmetic' | | Baffle | | | IMC | | | TrojanTO | | |
| DT | Hopp | 0.100 | 0.711 | 0.175 | 0.010 | 0.821 | 0.020 | 0.020 | 0.956 | 0.039 |
| | Half | 0.020 | 0.920 | 0.039 | 1.000 | 0.942 | 0.970 | 1.000 | 0.981 | 0.990 |
| | Walk | 0.000 | 0.858 | 0.000 | 0.940 | 0.961 | 0.950 | 1.000 | 0.908 | 0.951 |
| | Ant | 0.003 | 0.680 | 0.007 | 0.000 | 0.911 | 0.000 | 0.247 | 0.829 | 0.249 |
| | Kit | 0.953 | 0.635 | 0.746 | 0.843 | 0.452 | 0.588 | 0.987 | 0.418 | 0.586 |
| | Pen | 0.000 | 0.991 | 0.000 | 0.000 | 0.911 | 0.000 | 0.000 | 1.000 | 0.000 |
| GDT | Hopp | 0.060 | 0.703 | 0.111 | 0.000 | 1.000 | 0.000 | 0.130 | 0.693 | 0.219 |
| | Half | 0.000 | 1.000 | 0.000 | 0.010 | 1.000 | 0.020 | 0.907 | 1.000 | 0.946 |
| | Walk | 0.000 | 1.000 | 0.000 | 0.000 | 1.000 | 0.000 | 0.145 | 1.069 | 0.255 |
| | Ant | 0.000 | 0.633 | 0.000 | 0.000 | 1.000 | 0.000 | 0.003 | 0.908 | 0.007 |
| | Kit | 0.033 | 0.443 | 0.045 | 0.520 | 0.454 | 0.396 | 0.877 | 0.925 | 0.895 |
| | Pen | 0.000 | 0.841 | 0.000 | 0.000 | 1.000 | 0.000 | 0.000 | 1.000 | 0.000 |
| DC | Hopp | 0.600 | 0.790 | 0.675 | 0.403 | 0.766 | 0.512 | 0.927 | 0.861 | 0.893 |
| | Half | 0.000 | 0.924 | 0.000 | 0.370 | 0.940 | 0.464 | 1.000 | 1.000 | 1.000 |
| | Walk | 0.037 | 0.883 | 0.070 | 0.863 | 1.000 | 0.922 | 1.000 | 0.997 | 0.999 |
| | Ant | 0.010 | 0.884 | 0.020 | 0.467 | 0.860 | 0.518 | 0.667 | 0.903 | 0.637 |
| | Kit | 0.207 | 0.779 | 0.323 | 0.960 | 1.000 | 0.979 | 0.997 | 0.976 | 0.986 |
| | Pen | 0.000 | 0.897 | 0.000 | 0.000 | 0.978 | 0.000 | 0.000 | 1.000 | 0.000 |
| **Average** | | 0.112 | 0.809 | 0.123 | 0.355 | 0.889 | 0.352 | **0.550** | **0.912** | **0.536** |
| target action 'fixed random' | | Baffle | | | IMC | | | TrojanTO | | |
| DT | Hopp | 0.010 | 0.810 | 0.020 | 0.010 | 0.908 | 0.020 | 0.067 | 0.812 | 0.121 |
| | Half | 0.010 | 0.968 | 0.020 | 1.000 | 0.931 | 0.964 | 1.000 | 0.982 | 0.991 |
| | Walk | 0.000 | 0.885 | 0.000 | 0.290 | 0.949 | 0.444 | 0.970 | 0.898 | 0.933 |
| | Ant | 0.000 | 0.589 | 0.000 | 0.000 | 1.000 | 0.000 | 0.000 | 0.823 | 0.000 |
| | Kit | 0.883 | 0.692 | 0.766 | 0.990 | 0.426 | 0.595 | 0.923 | 0.459 | 0.604 |
| | Pen | 0.430 | 1.000 | 0.577 | 1.000 | 1.000 | 1.000 | 0.983 | 1.000 | 0.991 |
| GDT | Hopp | 0.060 | 0.614 | 0.109 | 0.020 | 0.803 | 0.039 | 0.395 | 0.774 | 0.382 |
| | Half | 0.000 | 1.000 | 0.000 | 0.850 | 1.000 | 0.919 | 0.993 | 1.000 | 0.997 |
| | Walk | 0.000 | 1.000 | 0.000 | 0.000 | 1.000 | 0.000 | 0.110 | 1.199 | 0.202 |
| | Ant | 0.000 | 0.554 | 0.000 | 0.000 | 1.000 | 0.000 | 0.000 | 0.980 | 0.000 |
| | Kit | 0.040 | 0.471 | 0.054 | 0.703 | 0.891 | 0.767 | 0.800 | 0.805 | 0.791 |
| | Pen | 0.022 | 0.852 | 0.042 | 0.795 | 0.747 | 0.769 | 1.000 | 0.927 | 0.959 |
| DC | Hopp | 0.137 | 0.753 | 0.214 | 0.647 | 0.661 | 0.649 | 0.867 | 0.815 | 0.835 |
| | Half | 0.000 | 1.000 | 0.000 | 0.427 | 1.000 | 0.579 | 1.000 | 1.000 | 1.000 |
| | Walk | 0.000 | 1.000 | 0.000 | 0.263 | 1.000 | 0.351 | 0.985 | 0.953 | 0.969 |
| | Ant | 0.010 | 0.689 | 0.020 | 0.693 | 0.811 | 0.742 | 0.053 | 0.842 | 0.091 |
| | Kit | 0.237 | 0.751 | 0.347 | 0.907 | 1.000 | 0.951 | 0.890 | 0.998 | 0.937 |
| | Pen | 0.549 | 0.928 | 0.687 | 0.970 | 0.959 | 0.964 | 0.283 | 0.951 | 0.431 |
| **Average** | | 0.133 | 0.809 | 0.159 | 0.531 | 0.894 | 0.542 | **0.629** | **0.901** | **0.624** |

