# OpenReview forum: "TrojanTO: Action-Level Backdoor Attacks Against Trajectory Optimization Models"
_ICLR.cc/2026/Conference — ICLR 2026 Poster_

### Official Review · Reviewer_f45h · 2025-10-29

**Soundness:** 3
**Presentation:** 3
**Contribution:** 3
**Rating:** 6
**Confidence:** 5

**Summary:**

The paper proposes the first method for backdoor attacks against Trajectory Optimization methods (TO). The authors first explore some challenges in attacks against TO models along with challenges in poisoning continuous action space RL in general. Using these motivations they develop their method, TrojTO, which iteratively optimizes the agent's policy along with the trigger such that the agent learns some desired backdoor attack behavior. This poisoning occurs at a post-training, fun-tuning phase over which the adversary has full control. They evaluate their method on MuJoCo tasks and compare to some related works on offline data poisoning.

**Strengths:**

* As far as I know, this is the first paper studying backdoor attacks against TO models.

* The proposed threat model is reasonable, though it isn't unique to TO models or RL in general.

* The paper properly places itself within the related work.

* I greatly appreciate sections 3.1 and 3.2 since the choice of target action and trigger design are often over looked in the study of backdoor attacks in general, especially against continuous control agents. The findings are also consistent with informal observations I have made while studying backdoor attacks.

* Section 3.3 does a good job of motivating the method.

* The proposed method is fairly simple but seems to work well.

* Evaluation is fairly comprehensive with many ablations in the appendix. The effort is appreciated.

**Weaknesses:**

1.) Even though the threat model is reasonable, I do think it is still fairly strong as the adversary has full control over the fine-tuning phase of the attack. Given this it would have been nice if the authors could devise an "optimal" attack which future attacks, with weaker threat models, could try to replicate.

2.) The paper spends a decent amount of time designing and figuring out what the optimal trigger is. In my opinion the trigger is a constraint of the test time environment and should therefore be determined based upon real world feasability. Therefore, I think this makes the authors' chosen evaluation environments somewhat mislead. In MuJoCo envionments one is unable to modify the agent's observations at test time to exploit the trigger without manipulating their internal, proprioceptive sensors. Therefore the trigger requires invasive levels of test time access to exploit. I think evaluating on MuJoCo environments like this is fine for evaluation purposes, but it would be insightful to additionally evaluate on image-based versions of MuJoCo since images are easier to manipulate at test time.

### Minor Weaknesses

* I understand that CP is the authors' chosen metric, but I would appreciate if they could simply underline the best performer in terms of ASR and BPT for each environment. To readers like myself CP is a difficult metric to interpret while ASR and BTP separately are easier to reason about. Therefore having underlines will make such a comparison much easier without compromising the authors' chosen metric.

* Missing table reference around line 448

* Typo in heading of main body section 2.4 and Appendix section C, should probably be "Experimental Setup"

**Questions:**

* Where does the adversary obtain the fine-tuning dataset they manipulate? (Important)

* Which part of the TrojTO method do you think is most important to its success?

* Would Baffle perform similarly to TrojTO if it also used trigger optimization?

* Why would an adversary choose to target and release a TO model rather than a standard RL model?

---

> ### Author Response · Authors · 2025-11-21
> **Thanks for your valuable feedback.**
>
> We sincerely thank you for your insightful comments, which have catalyzed numerous enhancements and refinements to the paper. In the following, we reply to the questions one by one for the convenience of checking.
>
> ---
> **W1**: An "optimal" attack.
>
> **A1**: Thanks for your valuable feedback. Our work focuses on revealing the vulnerability of TO models and elucidating the nature of backdoors in continuous action spaces.
>
> TrojanTO still requires additional training. A key direction for future work is investigating training-free backdoor methods (e.g., direct weight manipulation) in a continuous action space. This would allow for efficient backdoor injection, moving the field closer to an optimal, low-cost attack vector.
>
> ---
> **W2**: evaluate on image-based versions.
>
> **A2**: Thanks for your valuable feedback. We agree that extending to image-based domains is a valuable and natural step, particularly for exploring triggers with more direct real-world feasibility.
>
> Our current study focused on state-based environments to isolate the decision-making process from visual perception modules. Moreover, moving to image-based tasks is a non-trivial extension, requiring different model architectures and trigger design. We have incorporated this important direction into the discussion section to encourage further research.
>
> ---
> **MW1**: Underline the best ASR and BTP.
>
> **A3**: Thanks for your valuable feedback. We have revised all tables to boldface the best ASR and BTP values in the main body, as well as the best average values in the appendix.
>
> ---
> **MW2 & MW3**: Missing table reference and typos.
>
> **A4**: Thanks for your valuable feedback. We have added the missing table reference and corrected these typos in the revised manuscript.
>
> ---
> **Q1**: Where does the adversary obtain the fine-tuning dataset they manipulate?
>
> **A5**: Thanks for your valuable feedback. The adversary only needs a very small subset of clean trajectories (e.g., 0.3% of the dataset in our experiments) to prepare the attack. This data is readily accessible in realistic scenarios.
>
> 1. Interact with the environment using the victim TO model. Since only a small amount of data is needed, these interactions can be seen as routine testing. For instance, it is common practice to run a few test episodes after loading a model to ensure it has been loaded correctly. An adversary's data collection perfectly mimics this type of routine check, making it stealthy and practically undetectable.
>
> 2. The adversary can also use other agents on the same task to generate a small set of trajectories.
>
> In our experiments, we simulate this process by randomly sampling from the dataset. We will include a discussion in Appendix C.
>
> ---
> **Q2**: Which part do you think is most important?
>
> **A6**: Thanks for your valuable feedback. Empirically, batch poisoning is critical as it inherently preserves the sequential context. At inference time, the model inputs a clean history followed by a single triggered state. Standard data poisoning fails to replicate this specific context, leading to inconsistent learning signals. Batch poisoning addresses this by modifying only the last transition of a batch rather than the entire trajectory. Besides, the ablation study shows that removing this component results in the most significant drop in ASR.
>
> Conceptually, alternating optimization is equally important. In continuous action spaces, both the trigger pattern and the target action profoundly impact the backdoor efficacy. Consequently, trigger optimization and advanced alternating optimization are essential to ensure a successful attack.
>
> ---
> **Q3**: Would Baffle perform similarly to TrojTO if it also used trigger optimization?
>
> **A7**: Thanks for your valuable feedback. We conducted a new experiment.
>
> ||ASR/BTP/CP|
> |-|-|
> |Baffle|0.30/0.82/0.34|
> |Baffle w/ TO|0.72/0.88/0.77|
> |TrojanTO|**0.93**/**0.94**/**0.93**|
>
> Results show that applying trigger optimization boosts Baffle's performance, particularly increasing its ASR from 0.30 to 0.72. However, guided by the principle of consistent poisoning, TrojanTO still achieves significantly superior performance.
>
> ---
> **Q4**: Why target and release a TO model?
>
> **A8**: Thanks for your valuable feedback. An adversary would target a TO model for several reasons.
>
> 1. Sequence models, particularly transformers, have become the dominant architecture. It also became popular in offline RL [1]. Consequently, a successful attack on such architectures yields a significant impact.
> 2. Despite the surging popularity of TO models, their vulnerabilities remain largely unexplored compared to the relatively well-studied security of standard RL, leaving them exposed to unknown threats.
> 3. The prevalent "pre-train, download, and fine-tune" workflow offers a direct and highly scalable vector for supply chain attacks.
>
> ---
>
> [1] For instance, the seminal work DT (Decision Transformer: Reinforcement Learning via Sequence Modeling) has amassed over 2,400 citations.

---

> > ### Comment · Reviewer_f45h · 2025-11-24
> > **Response to Authors**
> >
> > Hello,
> >
> > Thank you for your detailed response and additional experiments. Overall I think the paper is sound and worthy of publication as a poster. I look forward to seeing further discussion with the other reviewers.
> >
> > Thank you,
> >
> > Reviewer f45h

---

### Official Review · Reviewer_d4Cw · 2025-10-30

**Soundness:** 3
**Presentation:** 3
**Contribution:** 3
**Rating:** 6
**Confidence:** 3

**Summary:**

This paper introduces TrojanTO, a post-training, action-level backdoor attack framework targeting Trajectory Optimization (TO) models used in offline reinforcement learning. While previous backdoor attacks in reinforcement learning largely focused on manipulating rewards during training, TrojanTO departs from this paradigm by conducting post-training attacks that directly modify pretrained TO models. The authors begin by analyzing the roles of actions, states, and rewards in determining the vulnerability of TO models and find that reward manipulation has negligible influence, while action–trigger coupling is the key factor governing attack success. Building upon these insights, TrojanTO incorporates three main mechanisms—trajectory filtering, batch poisoning, and alternating training—to achieve both high attack effectiveness and stealth. The framework is evaluated extensively across six D4RL benchmark tasks and three representative TO architectures (Decision Transformer, Graph Decision Transformer, and Decision ConvFormer). Results demonstrate that TrojanTO achieves superior composite performance (CP = 0.701) with only 0.3% data poisoning, significantly outperforming existing baselines such as Baffle and IMC. Overall, the paper provides the first systematic study of action-level, post-training backdoors in trajectory optimization models, revealing an underexplored and practically relevant threat vector in modern reinforcement learning systems.

**Strengths:**

The main strength of this paper lies in its novel problem formulation and practical significance. By shifting the focus from training-time to post-training attacks, TrojanTO highlights an emerging vulnerability relevant to supply-chain security in large pretrained RL models. The analysis of key contributing factors—action, state, and reward—is methodical and provides new insights into the structural vulnerabilities of TO models. Methodologically, TrojanTO’s design elegantly combines trajectory filtering to preserve benign performance, batch poisoning for trigger consistency, and alternating optimization for co-training the trigger and model. This composition demonstrates thoughtful engineering and strong empirical grounding. The experimental section is comprehensive and robust: evaluations span diverse environments (locomotion, navigation, manipulation) and architectures, with consistent improvements in attack success rate (ASR) and stealth (BTP). Extensive ablations further clarify the role of each module, providing transparency into design choices. The paper is clearly written, well-organized, and thoroughly supported by appendices, including detailed algorithms and hyperparameter descriptions that enhance reproducibility. In summary, TrojanTO represents a well-executed and relevant contribution that substantially advances understanding of backdoor threats in offline RL.

**Weaknesses:**

Despite its strong empirical performance, the work is somewhat limited in theoretical depth and scope of generalization. The core algorithmic components—particularly alternating optimization via MI-FGSM and trajectory filtering—are based on established techniques, and the paper lacks a unifying theoretical framework to quantify stealth–efficacy trade-offs or to analyze convergence guarantees. Additionally, the experimental evaluation, while extensive, remains confined to the D4RL suite, which primarily includes simulated continuous-control environments. Demonstrating TrojanTO on more realistic or high-dimensional embodied tasks (e.g., robot manipulation in MuJoCo or real-world datasets) would strengthen the practical impact. Another limitation is the absence of defense or detectability analysis. Since the authors motivate TrojanTO as a supply-chain threat, it would be valuable to assess how easily such attacks could be detected or mitigated by standard backdoor defenses (e.g., activation clustering or fine-tuning). Finally, some assumptions—such as full access to model weights and perfect control over trigger insertion—are strong and may not hold under more restricted adversarial conditions. These issues do not undermine the main contribution but suggest avenues for extending and deepening the work.

**Questions:**

Several clarifications would improve understanding of the work. First, how sensitive is TrojanTO’s success to the weighting parameter λ that balances the poisoned and clean losses, and how does it affect the trade-off between attack success and benign performance? Second, what are the computational and memory costs of alternating training relative to standard fine-tuning or model-editing attacks such as TrojanEdit (Guo et al., 2024b)? Third, can the authors discuss the attack’s robustness in black-box or transfer settings, where the attacker does not have full model access? Fourth, how does TrojanTO perform if the model undergoes partial fine-tuning or pruning after attack insertion—does the backdoor persist or degrade? Finally, are there practical countermeasures or defensive signals (e.g., gradient or activation anomalies) that defenders could exploit to identify TrojanTO-style backdoors during model validation?

---

> ### Author Response · Authors · 2025-11-21
> **Thanks for your valuable feedback.**
>
> We sincerely thank you for your insightful comments, which have catalyzed numerous enhancements and refinements to the paper. In the following, we reply to the questions one by one for the convenience of checking.
>
> ---
> **W1**: The core algorithmic components are based on established techniques.
>
> **A1**: Thanks for your valuable feedback. We wish to clarify that we introduce two novel components (trajectory filtering and batch poisoning) to overcome the challenges of consistency in backdoor implantation against TO models. To the best of our knowledge, we are the first to propose these methods in the area of RL backdoors.
>
> ---
> **W2**: More realistic or high-dimensional embodied tasks (e.g., robot manipulation in MuJoCo or real-world datasets) would strengthen the practical impact.
>
> **A2**: Thanks for your valuable feedback. Our evaluation includes standard control tasks and high-dimensional embodied tasks like Kitchen (59-dim obs., 9-dim act.) and Pen (45-dim obs., 24-dim act.) to demonstrate the practical impact on high-dimensional, embodied tasks. The attack's success in these settings shows its applicability is not confined to simple simulations.
>
> ---
> **W3 & Q4 & Q5**: Standard backdoor defenses.
>
> **A3**: Thanks for your valuable feedback. We conducted a comprehensive evaluation of several methods, including weight pruning, provable defense [1], spectral analysis, activation clustering, and fine-tuning. Our results indicate that while fine-tuning demonstrates effectiveness, **the other tested methods were largely unable to mitigate the TrojanTO attack**. The results are detailed in A1 to reviewer Q3rq and Appendix B.1 in the revised manuscript.
>
> ---
> **W4 & Q3**: Some assumptions—such as full access to model weights and perfect control over trigger insertion—are strong.
>
> **A4**: Thanks for your valuable feedback. We address the assumptions below.
>
> 1. Full access to model weights: TrojanTO is based on the post-training attack paradigm, which is a practical threat model [2,3]. In this setting, the core premise is that an adversary obtains a pre-trained model and then modifies its weights, the access is a definitional assumption of this threat model. This class of post-hoc vulnerability has been largely overlooked in RL.
>
> 2. Perfect control over trigger: We explicitly investigated the TrojanTO's robustness to noisy state observations, a common real-world factor. Results can be seen in Table 7 in the paper. What's more, trigger in TrojanTO is localized to a small subset of the state dimensions, therefore remains practical in real-world scenarios where complete state observation is not guaranteed.
>
> We hope this clarification adequately addresses your concerns.
>
> ---
> **Q1**: The weighting parameter $\lambda$.
>
> **A5**: Thanks for your valuable feedback. The ablation study on $\lambda$ shows that both ASR and BTP peak at $\lambda=10$ before declining. Consequently, we recommend a $\lambda$ value between 1 and 20.
>
> |Lambda|0.05|0.5|1|10|20|40|60|
> |-|-|-|-|-|-|-|-|
> |ASR|0.47|0.50|0.51|**0.98**|0.83|0.71|0.74|
> |BTP|0.63|0.62|0.72|**0.75**|0.68|0.59|0.45|
> |CP|0.46|0.45|0.48|**0.84**|0.70|0.58|0.56|
>
>
> **Q2**: The computational and memory costs of alternating training.
>
> **A6**: Thanks for your valuable feedback. TrojanTO involves two forward passes and one backward pass per step. Since the backward pass is the primary computational bottleneck, our cost remains comparable to standard fine-tuning. Besides, we believe the computation comparison to TrojanEdit is less informative because it is a pioneering data poisoning backdoor framework for the domain of image editing, while our work introduces a post-training attack for TO models.
>
> ---
>
> [1] Provable Defense against Backdoor Policies  in Reinforcement Learning.\
> [2] Attacks in Adversarial Machine Learning: A Systematic Survey from the Life-cycle Perspective.\
> [3] Machine Learning Models Have a Supply Chain Problem.

---

### Official Review · Reviewer_ozCU · 2025-10-31

**Soundness:** 3
**Presentation:** 3
**Contribution:** 2
**Rating:** 4
**Confidence:** 5

**Summary:**

The paper studies the problem of backdoor attacks against Trajectory optimization algorithms that try to predict the next actions based on partial trajectory and expected reward to go inputs. The authors find that the traditional method of reward manipulation backdoor attacks is not very effective for TO algorithms and instead proposes a fine-tuning method to inject backdoor using state and action manipulation. Their alternative optimization algorithm utilizes trajectory filtering with batch poisoning to install the backdoor in the policy. The authors present extensive experiments to support the effectiveness of their methods in various TO domains.

**Strengths:**

1. The paper is among the first few to study backdoor attacks in Trajectory Optimization settings to expose their vulnerability to backdoor attacks.
2. They propose an effective method called TrojanTO to successfully install a backdoor in pretrained policy through fine-tuning.
3. The authors have presented good experimental results to validate the effectiveness of their algorithm using dataset from different environments.

**Weaknesses:**

1. TO is nothing but a supervised learning problem where the input space is a sequence and output space is an action. As such, attacking a TO algorithm is same as attacking a supervised sequential model that has been studied a lot in the past. So, the correct comparison should be to compare it to a backdoor attack method in a supervised learning setting.
2. It is not surprising that reward manipulation does not lead to successful backdoors in TO because TO never tries to optimize for rewards so a reward signal cannot be used to install a bad behavior. Rather it takes rewards-to-go as input and only tries to predict an action that will achieve the corresponding reward input. So, direct action manipulation in the triggered state should be the way to go and there appears almost no value of doing an experiment to refute that the reward manipulation can be effective.
3. The paper also lacks a related works section. The authors have failed to articulate the novelty of their method compared to prior works and so it's difficult to position the contribution of this paper well.

**Questions:**

1. Why is attack on TO different from a standard supervised learning attack(say on sequential models)?
2. Have you done any comparisons with backdoor attack algorithms in supervised learning settings?

---

> ### Author Response · Authors · 2025-11-21
> **Thanks for your valuable feedback.**
>
> We sincerely thank you for your insightful comments, which have catalyzed numerous enhancements and refinements to the paper. In the following, we reply to the questions one by one for the convenience of checking.
>
> ---
> **W1 & Q1 & Q2**: Difference. & Compare TrojanTO to a backdoor attack method in an SL setting.
>
> **A1**: Thanks for your valuable feedback. We agree that TO models reframe the decision-making problem as a sequence modeling task. However, this formal similarity is deceptive. This is because the temporal nature of the data and the continuous regression objective present fundamentally different challenges compared to typical SL settings.
>
> 1. SL attacks typically operate on i.i.d. data, where simple poisoning is often sufficient. In contrast, TO models rely on temporal dependencies. Naively poisoning the dataset disrupts the sequential context of the trigger. For example, the baseline Baffle requires poisoning 10% of the dataset, but only achieves an ASR of 0.43.
>
> 2. Attacks are not trivially transferable across domains (e.g., CV to NLP), and RL is no exception. TO models perform regression in continuous action spaces based on sequences of states, actions, and rtg, which introduces unique challenges. Notably, we found that attack difficulty in RL is highly sensitive to the target action. While extreme target actions (e.g., ±1) are easier to force, precise non-extreme targets are significantly harder to implant.
>
> We empirically validate these differences using IMC [1], a representative SL attack which we carefully adapted for the TO setting. As shown in Table 4 in the paper, TrojanTO substantially outperforms IMC (CP 0.70 vs. 0.55).
>
> We further compare TrojanTO with LWP [2], an SL attack that poisons model weights layer by layer. The results below demonstrate that naively migrating SL attacks is ineffective. LWP's performance was severely limited, falling short of TrojanTO even when applied to all layers.
>
> ||ASR/BTP/CP|
> |-|-|
> |LWP w/ block1|0.00/0.83/0.00|
> |LWP w/ block2|0.00/0.80/0.00|
> |LWP w/ block3|0.01/0.65/0.03|
> |LWP w/ all|0.28/0.74/0.28|
> |TrojanTO|**1.00**/**0.97**/**0.99**|
>
> This gap confirms that typical SL attacks are limited by the inability to handle the temporal dependencies and continuous nature in RL. In contrast, TrojanTO secures significant performance gains through batch poisoning and trajectory filtering.
>
> ---
> **W2**: It is not surprising that reward manipulation does not lead to successful backdoors in TO.
>
> **A2**: Thanks for your valuable feedback. We agree that the ineffectiveness of reward manipulation for TO models is logical from a technical standpoint.
>
> However, reward manipulation is the dominant paradigm in RL backdoors (as rewards fundamentally distinguish RL from SL), and a reward-agnostic approach often causes confusion for those used to traditional RL settings, even in TO settings. Therefore,  to ensure completeness and prevent potential confusion, we conducted extensive experiments to empirically demonstrate this limitation across various TO models, including DT, DC, and GDT. By empirically showing that reward manipulation fails in TO, we confirm that manipulating rewards is ineffective, whereas optimizing the trigger and target action is critical for success.
>
> ---
> **W3**: Lacks a related works section.
>
> **A3**: We apologize for the confusion regarding the related works section. It was initially placed in Appendix A due to page limits. In the revised manuscript, we have addressed this by **moving the core related work on RL backdoors back into the main body**. A broader introduction of the post-training attack paradigm and its potential scenarios in RL is now detailed in Appendix A. We hope this new structure better highlights the novelty of our work and addresses your concern fully.
>
> Furthermore, to clearly articulate our contributions in contrast to prior art:
>
> 1. To the best of our knowledge, this is the first work to propose a backdoor attack targeting TO models. We reveal that this important and increasingly popular class of models is highly vulnerable to post-training attacks.
>
> 2. We conduct a systematic empirical study on how the core elements of RL (state, action, and reward) influence backdoors in continuous action spaces. As Reviewer f45h insightfully noted, aspects like trigger design and target action selection are often overlooked. Our work fills this gap by providing a detailed analysis for designing such attacks.
>
> 3. Building on the principle of consistent poisoning, TrojanTO uniquely integrates trajectory filtering and batch poisoning. Extensive experiments demonstrate the effectiveness of TrojanTO across a variety of RL tasks and TO models, evaluated in scenarios involving diverse target actions.
>
> ---
> [1] A Tale of Evil Twins: Adversarial Inputs versus Poisoned Models.\
> [2] Backdoor Attacks on Pre-trained Models by Layerwise Weight Poisoning.

---

### Official Review · Reviewer_Q3rq · 2025-11-01

**Soundness:** 3
**Presentation:** 3
**Contribution:** 3
**Rating:** 6
**Confidence:** 5

**Summary:**

This paper investigates the security vulnerabilities of Trajectory Optimization (TO) models in offline reinforcement learning (RL) and introduces TrojanTO, the first action-level backdoor attack targeting such models. While backdoor attacks have been well-studied in traditional RL agents (which rely on reward-driven policy updates), the paper argues that these methods are largely ineffective against TO models such as Decision Transformer (DT), Goal Decision Transformer (GDT), and Decision ConvFormer (DC), due to their sequence modeling nature (transformer-based trajectory modeling rather than reward-based learning), large network capacity, and continuous action spaces that complicate precise manipulation. TrojanTO is proposed as a post-training attack that forges strong couplings between triggers and target actions without interfering with the training pipeline.  Experiments show that TrojanTO achieves high attack success rates (ASR) across multiple TO model architectures and tasks while maintaining performance on benign data

**Strengths:**

The TrojanTO design consists of three key components: Alternating Training - reinforces the association between the trigger pattern and target action; Trajectory Filtering - maintains benign performance by filtering non-critical trajectories; and Batch Poisoning - ensures trigger consistency and stealthiness across evaluation conditions. This novel combination of ideas makes TrojanTO effective and data-efficient for injecting post-training, action-level backdoors in offline trajectory optimization models.

The paper conducts a systematic empirical study of how action, state, and reward manipulations affect TO models, leading to the finding that reward manipulation is largely ineffective while action-state triggers dominate attack success.

The paper has strong empirical evaluation and the attack requires very small poisoning budget.

**Weaknesses:**

The paper does not discuss possible defenses or mitigation strategies against TrojanTO. Even a brief evaluation of trigger detectability, model auditing, or defensive retraining would help position this research within the broader context of AI security and trustworthiness. The appendix B.1 has some discussion on defense but a more focuesed synopsis of experiments could be added to the main paper.

**Questions:**

Have you tested whether methods such as fine-tuning, weight pruning, spectral analysis, or trigger anomaly detection can mitigate TrojanTO? Some of these are very briefly mentioned in B.1 appendix but it is unclear if there was a statistically significant observation made in the experiments.

Could TrojanTO be executed on large foundation-scale TO models (e.g., Gato, RT-2)?

---

> ### Author Response · Authors · 2025-11-21
> **Thanks for your valuable feedback.**
>
> We sincerely thank you for your insightful comments, which have catalyzed numerous enhancements and refinements to the paper. In the following, we reply to the questions one by one for the convenience of checking.
>
> ---
> **W1 & Q1**: Possible defenses or mitigation strategies against TrojanTO.
>
> **A1**: Thanks for your valuable feedback. We additionally conducted an evaluation of several methods, including **weight pruning, provable defense [1], spectral analysis, in addition to activation clustering and fine-tuning**. Our results demonstrate that fine-tuning is the most effective defense, while the other tested methods were largely unable to mitigate the TrojanTO attack.
>
> 1. Weight pruning: Our evaluation of weight pruning using both activation-based and magnitude-based methods. Our experiments show that simple weight pruning is a poor trade-off, as it sacrifices significant BTP for incomplete and unreliable backdoor removal.
>
>     |Method|Ratio|ASR|BTP|
>     |-|-|-|-|
>     |Activation|0.03|1.00→0.67|0.85→0.58|
>     ||0.15|1.00→0.44|0.69→0.21|
>     |Magnitude|0.02|1.00→1.00|0.87→0.80|
>     ||0.15|1.00→1.00|0.81→0.34|
>
> 2. Provable defense [1]: This defense purifies triggers by projecting states into a safe subspace. However, it causes the BTP to range from 0.80 to 0.16. This suggests that the projection distorts or discards essential state features required for the benign task, rendering the agent ineffective.
>
>     |project_dim|ASR|BTP|
>     |-|-|-|
>     |8|1→0.00|0.80→0.08|
>     |10|1→0.19|0.80→0.16|
>
> 3. Spectral analysis: The spectral analysis detector is impractical due to the true positive rate (TPR) exhibiting significant variance across different environments. Furthermore, we find that TPR is highly sensitive to its hyperparameters (see Appendix B in the revised manuscript for detailed results).
>
>     |Env|TPR|FPR|
>     |-|-|-|
>     |Hopp|0.96|0.13|
>     |Half|0.00|0.05|
>     |Walk|0.59|0.12|
>
> 4. Activation clustering: As demonstrated in Appendix B.1, activation clustering fails to detect the backdoor because the internal activation space is dominated by the significant variations between different input modalities (e.g., states, actions, return-to-go).
>
> 5. Fine-tuning: Fine-tuning serves as an effective defense that suppresses the backdoor attack. As shown in the table below, using only 10\% of the original training steps, it reduces the ASR to near zero while maintaining the BTP. However, while it successfully reduces the ASR, a residual effect from the attack persists, as the output action's continued proximity to the target action.
>
>     |Steps|1000|3000|5000|7000|9000|
>     |-|-|-|-|-|-|
>     |BTP|0.99|0.98|0.95|0.98|0.93|
>     |ASR|1.00|1.00|1.00|0.76|0.28|
>
> We will include both the new defense evaluations and the results across various hyperparameters in our revised manuscript. The full details will be presented in Appendix B.1, and a new Section 6.5 will be added to the main paper. Thanks again for your valuable feedback.
>
> ---
> **Q2**: Could TrojanTO be executed on large foundation-scale TO models (e.g., Gato, RT-2)?
>
> **A2**: Thanks for your valuable feedback. The principles of TrojanTO are general, and we believe they can be extended to large foundation-scale TO models. However, Gato is not publicly available, and there are no complete community reproductions, posing a practical challenge. RT-2, on the other hand, employs a Vision-Language-Action architecture, which differs significantly from the TO models we focus on, as its inputs include images and text.
>
> A more suitable candidate for this evaluation would be LaMo [2], which leverages a pre-trained GPT-2 (200M) and is compatible with RL tasks. While we agree that this is an important and exciting direction, a thorough investigation of foundation-scale models constitutes a significant research project in its own right, as it needs substantial computational resources. Therefore, this investigation is best positioned as a dedicated future study, rather than an addition to the current work.
>
> ---
> [1] Provable Defense against Backdoor Policies  in Reinforcement Learning.\
> [2] Unleashing the Power of Pre-trained Language Models for Offline Reinforcement Learning.

---

> > ### Comment · Reviewer_Q3rq · 2025-11-27
> > **Thank you**
> >
> > Thanks. Yes, investigation extension to large models would be a significant undertaking.

---

### Author Response · Authors · 2025-11-22
**Gratitude to all Area Chairs and Reviewers**

Dear Area Chairs,

We are deeply grateful to you and the reviewers for your time and dedication, particularly in light of the recent unforeseen event. The constructive feedback provided has been instrumental in strengthening the technical completeness and clarity of our manuscript.

We summarize the key improvements below. All changes are highlighted in blue in the revised submission.

* We include three new defense evaluations in Appendix B.1, with a brief discussion provided in the newly added Section 6.5 of the main paper to further discuss these defensive methods.
* We moved the core related work from Appendix A back to Section 2 to ensure the paper is self-contained and easier to follow.
* We expanded the discussion on future image-based attacks in the Limitations section and clarified the data sources used by the adversary in Appendix C.1.
* We thoroughly polished the entire manuscript to correct grammatical errors and improve the overall presentation.

We conclude by reiterating our thanks to the Area Chairs and reviewers for your invaluable efforts and consideration.

---

### Meta-Review · Area_Chair_VJZU · 2026-01-08

**Summary:**

The submission studies the whitebox backdoor attacks for trajectory optimization models (such as decision transformers and decision convformers). By revealing key factors (target action, trigger design) and discarding negligible factors (reward design), the proposed post-training attack, TrojanTO, achieves strong performance.

Reviewers appreciate the novelty and strong empirical performance of the work. Here are key reviewers' concerns:
1. (shared concern) Lack of discussion on possible defenses or mitigations.
2. (shared concern) The threat model is relatively strong. Specifically, the trigger design may need to consider the real-world feasibility.
3. Lack of evaluation on large foundation-scale TO models, or evaluation on large-scale environments such as MuJoCo.
4. Lack of related work discussion.
5. Limited in theoretical depth and scope of generalization, or connection with other regimes such as supervised learning backdoors.

**Reviewer Concerns:**

The authors did a great job in rebuttal and preparing the paper revision. Two reviewers replied and acknowledged the rebuttal. Overall, I think most concerns are largely addressed, especially Concerns 1 and 4. The others are partly addressed but in my opinion the concerns are not critical (Concerns 2, 3 and 5). On the other hand, these two concerns, especially Concerns 2 and 3, point out potential future directions as they themselves require substantial effort and bring new findings.

**Reviewer Scores:**

Overall, after the discussion phase, I would anticipate that the scores may not change or may slightly increase. Two reviewers who replied did not indicate their score change. The other two reviewers may keep or slightly increase the score: Reviewer d4Cw may keep the score 6 as their concerns are mostly addressed but some remain. Reviewer ozCU may keep the score 4 or slightly increase to 6. Their concerns are more like a subjective judgment of the overall contribution. Rebuttal is not targeted for such concerns. But the reviewer's judgement may become more positive after reading other's reviews.

---

### Decision · Program_Chairs · 2026-01-26

Accept (Poster)